# Anchored Policy Optimization: Mitigating Exploration Collapse via Support-Constrained Rectification

**Tianyi Wang** [1][2][*]  **Long Li** [3][*]  **Hongcan Guo** [2]  **Yibiao Chen** [2]  **Yixia Li** [1]
**Yong Wang** [4]  **Yun Chen** [5]  **Guanhua Chen** [1][†]

## Abstract

Reinforcement Learning with Verifiable Rewards (RLVR) is increasingly viewed as a tree pruning mechanism. However, we identify a systemic pathology termed Recursive Space Contraction (RSC), an irreversible collapse driven by the combined dynamics of positive sharpening and negative squeezing, where the sampling probability of valid alternatives vanishes. While Kullback-Leibler (KL) regularization aims to mitigate this, it imposes a rigid Shape Matching constraint that forces the policy to mimic the reference model's full density, creating a gradient conflict with the sharpening required for correctness. We propose Anchored Policy Optimization (APO), shifting the paradigm from global Shape Matching to Support Coverage. By defining a Safe Manifold based on the reference model's high-confidence support, APO permits aggressive sharpening for efficiency while selectively invoking a restorative force during error correction to prevent collapse. We theoretically derive that APO serves as a gradient-aligned mechanism to maximize support coverage, enabling an Elastic Recovery that re-inflates valid branches. Empirical evaluations on mathematical benchmarks demonstrate that APO breaks the accuracy-diversity trade-off, significantly improving Pass@1 while restoring the Pass@K diversity typically lost by standard policy gradient methods.

[*]Equal contribution  [1]Southern University of Science and Technology [2]Beijing University of Posts and Telecommunications [3]Griffith University [4]Alibaba Group [5]Shanghai University of Finance and Economics. Correspondence to: Guanhua Chen <chengh3@sustech.edu.cn>.

*Proceedings of the $43^{rd}$ International Conference on Machine Learning*, Seoul, South Korea. PMLR 306, 2026. Copyright 2026 by the author(s).

## 1. Introduction

Reinforcement Learning with Verifiable Rewards (RLVR) has established itself as a standard paradigm for enhancing the reasoning capabilities of Large Language Models (LLMs) (Guo et al., 2025; Wen et al., 2025; Su et al., 2025). Recent theoretical insights, notably by , suggest that RLVR functions primarily as a "Tree Pruning"(Yue et al., 2025; Wang et al., 2025a) mechanism rather than synthesizing knowledge *ex nihilo*. In this view, optimization identifies and amplifies correct reasoning paths latent in the pretrained graph while suppressing incorrect branches. Under this hypothesis, the Reference Model serves as the primary source of potential validity.

Guided by the "Tree Pruning" perspective, we identify a systemic pathology termed **Recursive Space Contraction (RSC)** (Section 3.1). We find that pruning is often irreversible due to a dual dynamic: positive updates passively suppress alternative branches via sharpening, while negative updates blindly redistribute mass back to dominant tokens—a "rich-get-richer" effect (Ren & Sutherland, 2025) that fails to recover suppressed valid paths. Consequently, the policy locks into narrow, over-confident tracks, losing the probability mass needed for exploration. As shown in Figure 1(a), RSC creates a self-reinforcing loop. While methods like Negative Sample Reinforcement (NSR) (Zhu et al., 2025) aim to maintain diversity, they merely modulate update intensity without addressing the blind nature of probability redistribution inherent in negative feedback. Without a constructive recovery mechanism, the loss of the reference model's valid support becomes permanent (Brekelmans et al., 2022; Shumailov et al., 2024; Huang et al., 2025a).

The challenge then lies in how to regulate this pruning process to allow for both efficient sharpening and robust recovery. The standard approach employs Kullback-Leibler (KL) regularization(Ouyang et al., 2022; Peters et al., 2010), but we argue this represents a strictly *passive utilization* of the reference model, imposing a rigid Shape Matching constraint (Section 3.3) (Minka et al., 2005). This forces the policy model to mimic the reference model's full distribution, including its noise, creating a fundamental Gradient

Conflict. As illustrated in Figure 2, the KL penalty generates a counter-gradient that often competes with the reward signal; this conflict can drive the resultant update direction to diverge from the optimization Trust Region(Schulman et al., 2017a), leading to instability or suboptimal convergence. As derived in Section 4, to improve efficiency (Pass@1), the policy must sharpen around correct paths, yet the KL constraint compels the policy to maintain a broader uncertainty profile. This conflict is particularly detrimental: it hinders necessary sharpening during correct reasoning while failing to provide a proactive direction for manifold restoration during error correction.

To resolve the RSC problem and the resulting Gradient Conflict, we introduce **Anchored Policy Optimization (APO)**. Shifting the paradigm from Shape Matching to Support Coverage, APO defines a **Safe Manifold**—the high-confidence support set of the reference model. We propose that regularization should be active and conditional: valid sharpening is permitted to maximize efficiency, but encountering an error triggers a restorative force toward the Safe Manifold. Crucially, APO resolves the Gradient Conflict through Ratio Rectification. By injecting a pulling force directly into the policy ratio, APO generates gradients that are intrinsically aligned with the reward signal, facilitating an **Elastic Recovery** that re-inflates neglected valid branches without the rigid overhead of density matching (Figure 1(c)).

We substantiate this approach through theoretical analysis, demonstrating that the APO rectification term functions as a gradient-aligned proxy for maximizing Support Coverage. We derive that our update rule produces gradients strictly collinear with maximizing the total mass of the Reference Model's Safe Manifold during error correction, providing a formal guarantee for the recovery mechanism detailed in Section 4. Empirical evaluations on five mathematical benchmarks—including **AIME 24/25**, **Math500**, and **Minerva**—confirm that APO breaks the accuracy-diversity trade-off, outperforming KL-based baselines by up to **6%** in $Pass@1$ efficiency while recovering **1.5%–3.3%** in $Pass@K$ diversity typically lost in standard policy gradient methods[1].

## 2. Preliminaries

### 2.1. RLVR and the Shape Constraint

We focus on Reinforcement Learning with Verifiable Rewards (RLVR) using algorithms like PPO (Schulman et al., 2017b) and GRPO (Shao et al., 2024). These methods optimize a clipped surrogate objective to maintain policy stability. Letting $r_t(\theta) = \frac{\pi_\theta(y_t|x,y_{<t})}{\pi_{\text{old}}(y_t|x,y_{<t})}$ denote the probability ratio and $A_t$ the estimated advantage, the objective

is:

$$\mathcal{L}(\theta) = \mathbb{E}\left[\min\left(r_t(\theta)A_t, \text{clip}(r_t(\theta), 1-\epsilon, 1+\epsilon)A_t\right)\right].$$
(1)

Standard approaches augment this with a KL penalty, $D_{\text{KL}}(\pi_\theta \| \pi_{\text{ref}})$. We define this as a **Shape Constraint**: minimizing KL imposes a global density matching requirement. This forces $\pi_\theta$ to mimic the reference model's full distribution profile—including noise—rather than solely retaining its knowledge support, hindering the sharpening required for Pass@1 efficiency.

### 2.2. The Squeezing Effect

Negative feedback on Softmax-parameterized policies induces a pathological dynamic termed the *Squeezing Effect* (Ren & Sutherland, 2025). Consider minimizing the probability of an error token $y_{\text{err}}$. The gradient update for the logit $z_k$ of any alternative token $k$ ($k \neq y_{\text{err}}$) follows:

$$\Delta z_k \propto -\nabla_{z_k}\pi_\theta(y_{\text{err}}) \propto \pi_\theta(y_{\text{err}}) \cdot \pi_\theta(k).$$
(2)

This reveals a *Rich-get-Richer* dynamic (Ren & Sutherland, 2025): the update magnitude for a token is strictly proportional to its current probability $\pi_\theta(k)$. Unlike linear redistribution, this mechanism disproportionately boosts the dominant path (where $\pi_\theta$ is large) while providing negligible updates to lower-probability valid branches. Consequently, negative feedback accelerates an irreversible collapse into low-entropy states, effectively "starving" the exploration of diverse reasoning paths.

## 3. Theoretical Framework

In this section, we provide a rigorous analysis of the optimization dynamics in RLVR. We first analyze why the Squeezing Effect (defined in Section 2.2) leads to irreversible pruning in reasoning tasks. Subsequently, we frame the regularization problem through a geometric lens, contrasting the restrictive *Shape Matching* paradigm of KL divergence with our proposed *Support Coverage* objective.

### 3.1. The Dual Dynamics of Recursive Space Contraction

While Ren & Sutherland (2025) identified the *Squeezing Effect* in offline settings, we define **Recursive Space Contraction (RSC)** as a systemic pathology in on-policy RL driven by the interplay of positive and negative feedback. Unlike offline frameworks with fixed data, on-policy optimization relies on stochastic sampling, creating a self-reinforcing loop: as the probability of valid alternatives declines, their sampling frequency vanishes, severing the gradient signal required for recovery.

Crucially, RSC is a compound result of dual update dynamics. During positive updates (sharpening), increasing

---

[1]Our code is available at https://github.com/1BIMU/APO_OFFICAL.

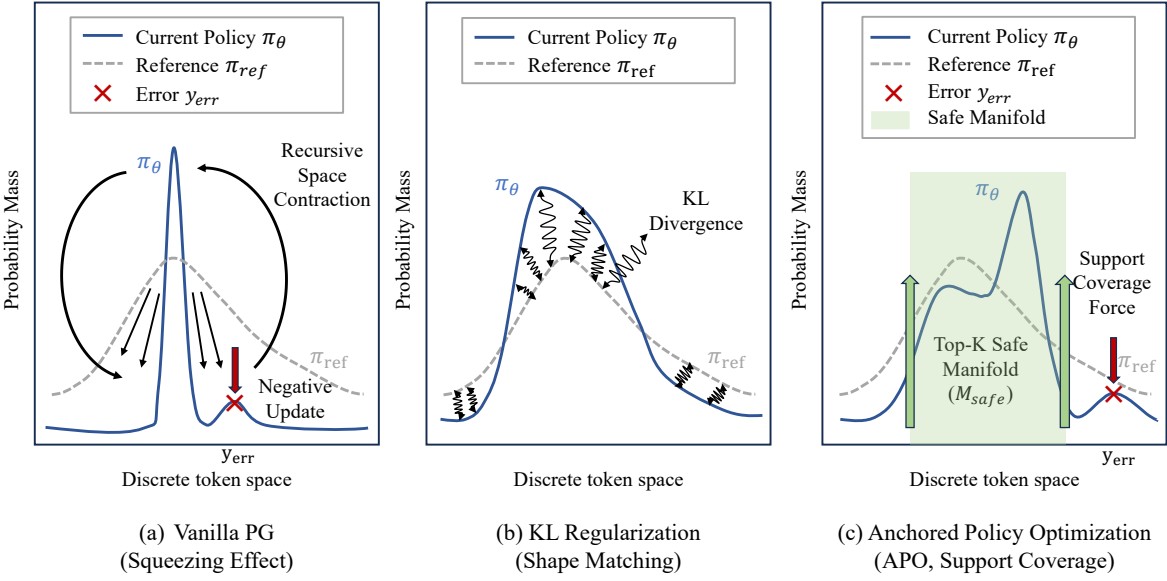

*Figure 1.* **Geometric Interpretation of Regularization Paradigms. (a) Recursive Space Contraction (Vanilla PG):** The interplay of positive sharpening and blind negative squeezing drives probability mass to collapse into a single narrow path, permanently discarding valid support. **(b) Shape Matching (KL Regularization):** KL imposes a rigid constraint (visualized as springs) that forces $\pi_\theta$ to mimic the exact density profile of $\pi_{\text{ref}}$, prohibiting the local sharpening necessary for high Pass@1 efficiency. **(c) Support Coverage (APO, Ours):** Our method maximizes mass coverage within a Safe Manifold ($\mathcal{M}_{\text{safe}}$). This permits aggressive sharpening for accuracy, while the anchor force provides **Elastic Recovery** during error correction to prevent leakage into invalid regions.

the probability of a correct path passively suppresses all unsampled tokens—including valid alternatives—via partition function inflation. Critically, subsequent negative updates fail to reverse this: the *Squeezing Effect* blindly redistributes the released mass to already dominant tokens rather than re-inflating the suppressed tail. Consequently, once a valid reasoning path drifts into the "tail", it becomes mathematically unreachable regardless of its validity (see formal derivation in Appendix B.2). This results in an **irreversible** collapse, where the policy permanently discards latent reasoning capabilities, as visualized in Figure 3.

*Table 1.* **Oracle Coverage Analysis (Motivation).** Standard RLVR tends to collapse distribution towards the path (Top-1). However, our analysis reveals that the path only captures 83.84% of correct reasoning steps. In contrast, the *Safe Manifold* defined by Top-8 covers 97.47%. This implies that blind squeezing is mathematically guaranteed to eliminate ∼16% of valid reasoning paths, necessitating a support-constrained approach.

| Safe Manifold Scope | Recall (Coverage) | Loss Rate |
|---|---|---|
| **The Path (Top-1)** | 83.84% | **16.16%** |
| Top-4 | 95.52% | 4.48% |
| Top-8 | **97.47%** | **2.53%** |
| Top-16 | 98.46% | 1.54% |

### 3.2. Empirical Motivation: The Poverty of the Path

The concern regarding the RSC raises a critical question: If RL collapses the policy onto the single most probable path, does collapsing onto the most likely path force the model to ignore other correct ways to solve the problem?

To quantify this, we analyze the oracle coverage rate of Qwen2.5-Math-7B on 1,000 randomly sampled MATH(Hendrycks et al., 2021) problems. We employ a teacher-forcing protocol, feeding ground-truth prefixes $y^*_{<t}$ to compute next-token logits. We measure *Recall*: the probability that the actual ground truth token $y^*_t$ falls within the Top-$K$ prediction set: $\mathbb{P}(y^*_t \in \text{TopK}(\pi_{\text{ref}}(\cdot|x, y^*_{<t})))$.

Table 1 reveals a critical insight: strictly trusting the greedy path (Top-1) incurs a substantial **16.16% Loss Rate** of valid tokens. This quantitative evidence exposes a fundamental flaw in the prevailing "Tree Pruning" hypothesis, suggesting that optimizing exclusively for the single best path reduces not just solution diversity, but fundamental *validity*. Conversely, a modest expansion of the scope to the Top-8 recovers nearly the entire validity space (97.47% coverage) while preserving a computationally sparse search space. This significant empirical gap—between the poverty of the single path and the richness of its local support—serves as the primary motivation for our transition from rigid Shape Matching to targeted Support Coverage within the Top-8 Safe Manifold.

### 3.3. From Shape Matching to Support Coverage

To mitigate the aforementioned collapse, we categorize existing solutions and our proposed method based on the geometric nature of their constraints.

**Paradigm A: Shape Matching (Standard KL Regularization).** Standard RLVR utilizes the Kullback-Leibler divergence $D_{\mathrm{KL}}(\pi_\theta \| \pi_{\mathrm{ref}})$ as a penalty. Minimizing this divergence forces the policy $\pi_\theta$ to align its entire probability density function with $\pi_{\mathrm{ref}}$. Mathematically, this imposes a *global* constraint:

$$\pi_\theta(y) \approx \pi_{\mathrm{ref}}(y), \quad \forall y \in \mathcal{V}. \tag{3}$$

We term this *Shape Matching* because it compels the student model to mimic the exact "shape" of the teacher's distribution, including its uncertainties. While robust, this constraint is agnostic to the correctness of the reasoning path; it penalizes the RL agent for becoming confident (sharpening) even on correct answers.

**Paradigm B: Support Coverage (Anchored Policy Optimization).** We propose a relaxation of the global shape constraint, focusing instead on the geometric support. We define the *Safe Manifold* $\mathcal{M}_{\mathrm{safe}}$ as the high-confidence support set of the reference model:

$$\mathcal{M}_{\mathrm{safe}} = \{y \in \mathcal{V} \mid y \in \mathrm{TopK}(\pi_{\mathrm{ref}})\}. \tag{4}$$

Ideally, we seek to maximize the *Support Coverage Objective*:

$$\mathcal{J}_{\mathrm{support}}(\theta) = \sum_{y \in \mathcal{M}_{\mathrm{safe}}} \pi_\theta(y). \tag{5}$$

Unlike KL, which acts as a continuous penalty pulling the policy towards the reference center, Support Coverage acts as a boundary condition. It allows the policy to distribute mass freely *within* $\mathcal{M}_{\mathrm{safe}}$ (enabling sharpening for Pass@1) but requires a restoring mechanism when mass leaks to the unsafe region $\mathcal{V} \setminus \mathcal{M}_{\mathrm{safe}}$.

While this restricts the optimization to the reference's prior support, we argue that paths outside this manifold are practically unreachable via sampling in large-scale action spaces (see Appendix A.3 for a detailed discussion on reachability).

However, directly optimizing $\mathcal{J}_{\mathrm{support}}$ requires explicit auxiliary losses introduces a **Gradient Conflict**. As illustrated in Figure 2(a), an auxiliary loss term creates an orthogonal gradient component that competes with the reward signal, potentially dragging the update step outside the PPO trust region (see Appendix B.4 for a stability analysis). In the following Section 4, we introduce APO as a gradient-aligned modification (Figure 2(b)). Instead of adding an external force, APO injects a restoring force strictly collinear with $\nabla \mathcal{J}_{\mathrm{support}}$ intrinsic to the policy ratio, ensuring stable updates when negative feedback signals a potential collapse.

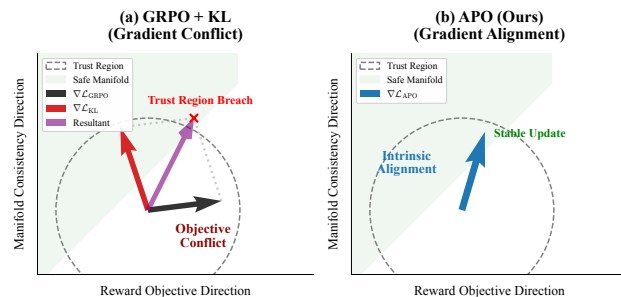

*Figure 2.* **Geometric Interpretation of Gradient Dynamics.** The x and y axes represent the directions of reward maximization and manifold consistency, respectively. **(a) GRPO + KL (Baseline):** The KL penalty gradient ($\nabla \mathcal{L}_{\mathrm{KL}}$, Red) conflicts with the reward gradient ($\nabla \mathcal{L}_{\mathrm{GRPO}}$, Black), driving the resultant vector (Purple) to breach the Trust Region. **(b) APO (Ours):** APO produces a unified gradient ($\nabla \mathcal{L}_{\mathrm{APO}}$, Blue) intrinsically aligned with the Safe Manifold, ensuring stable and efficient updates.

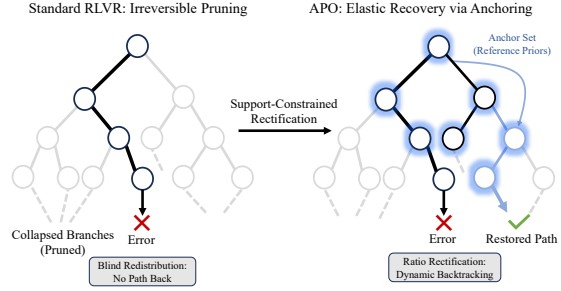

*Figure 3.* **Comparison of Error Correction Dynamics. (Left) Irreversible Pruning:** In Standard RLVR, the *Squeezing Effect* causes blind mass redistribution upon error, permanently collapsing alternative branches (greyed out). **(Right) Elastic Recovery via APO:** When the sharpened path hits an error (Red X), the **Pull Term** in our rectified ratio acts as a restoring force. It re-activates the *Anchor Set* (glowing blue nodes) derived from the reference priors, allowing the agent to "backtrack" and explore valid alternative paths (solid blue arrow) that were previously suppressed.

## 4. Method

To dismantle the **Recursive Space Contraction** loop, we adopt a targeted intervention strategy. We accept the *Sharpening* induced by positive updates as essential for sampling efficiency (Pass@1); therefore, we focus exclusively on rectifying the destruction caused by negative updates.

Standard RLVR algorithms operate via strict penalization: when an error occurs, they suppress the generated path but provide no guidance on alternative directions. This leads to the *Squeezing Effect* (Ren & Sutherland, 2025), where probability mass is blindly redistributed to the already dominant tokens, causing the distribution to collapse rather than recovering valid alternatives.

We introduce **Anchored Policy Optimization (APO)** to transform this irreversible pruning into an **Elastic Recovery** process. APO employs a dual-force mechanism: a

Push Force to suppress specific errors, and a corresponding Pull Force that actively re-inflates the *Safe Manifold* of the reference model, ensuring the policy backtracks to valid reasoning paths.

### 4.1. The Virtual Anchor Ratio and Exclusive Anchoring

To implement the Pull Force, we construct a proxy for the Safe Manifold's mass. However, a naive maximization of the reference support creates a **Signal Cancellation**. Since the error token $y_t$ is often high-probability in the reference model (i.e., $y_t \in \text{TopK}(\pi_{\text{ref}})$), including it in the anchor set would cause the optimizer to simultaneously suppress $y_t$ (via the advantage term) and boost $y_t$ (via the anchor term).We provide the analysis of gradient conflicts in Appendix D.1.

To resolve this, we impose **Exclusive Anchoring**. We define the *Anchor Set* $S_{\text{anchor}}$ as the Top-K reference tokens, *strictly excluding* the current error token:

$$S_{\text{anchor}} = \text{TopK}(\pi_{\text{ref}}(\cdot|x)) \setminus \{y_t\}. \quad (6)$$

The **Virtual Anchor Ratio** is defined to measure the policy's coverage of the safe manifold relative to the reference model. Let $Z_{\text{ref}}$ denote the total probability mass of the anchor set under the reference distribution, serving as the normalization constant:

$$Z_{\text{ref}} = \sum_{j \in S_{\text{anchor}}} \pi_{\text{ref}}(j). \quad (7)$$

We define the normalized importance weight $\hat{\omega}_k = \pi_{\text{ref}}(k)/Z_{\text{ref}}$ for each token $k \in S_{\text{anchor}}$. The anchor ratio $r_{\text{anchor}}$ is then constructed via importance sampling:

$$
\begin{aligned}
r_{\text{anchor}} &= \sum_{k \in S_{\text{anchor}}} \hat{\omega}_k \cdot \frac{\pi_\theta(k)}{\pi_{\text{ref}}(k)} \\
&= \sum_{k \in S_{\text{anchor}}} \frac{\pi_{\text{ref}}(k)}{Z_{\text{ref}}} \cdot \frac{\pi_\theta(k)}{\pi_{\text{ref}}(k)} \quad (8) \\
&= \frac{1}{Z_{\text{ref}}} \sum_{k \in S_{\text{anchor}}} \pi_\theta(k).
\end{aligned}
$$

This derivation explicitly shows that $r_{\text{anchor}}$ is proportional to the total probability mass $\pi_\theta$ assigns to the safe manifold, scaled by the inverse of the reference model's confidence $Z_{\text{ref}}$. This ensures $r_{\text{anchor}}$ acts as a clean, differentiable estimate of the *alternative* safe mass, strictly orthogonal to the error token.

### 4.2. Ratio Rectification

We integrate this anchor into the PPO objective via **Ratio Rectification**. For a negative advantage sample ($A_t < 0$),

we construct the rectified ratio $\tilde{r}_{\text{APO}}$:

$$\tilde{r}_{\text{APO}} = \underbrace{\lambda \cdot \frac{\pi_\theta(y_t)}{\pi_{\text{old}}(y_t)}}_{\text{Push: Suppress Error}} - \underbrace{\beta \cdot r_{\text{anchor}}}_{\text{Pull: Recover Support}}. \quad (9)$$

By minimizing $\tilde{r}_{\text{APO}}$, the optimizer essentially minimizes the difference between the error probability and the safe manifold probability. Appendix B.1 proves our pull gradient is collinear with the Support Coverage objective, ensuring geometric consistency.

Crucially, our theoretical derivation in Appendix B.3 proves that this pull force facilitates **Elastic Recovery**. Unlike uniform regularization, the restoring gradient scales proportionally with the policy's current confidence ($\nabla \propto \pi_\theta$), ensuring that the relative topological importance of valid candidates is preserved during recovery.

**Computational Efficiency.** Unlike KL divergence, which requires dense gradient propagation over the full vocabulary ($V \approx 128k$), APO operates on a sparse manifold ($K \ll V$, typically $K = 8$). This significantly reduces backward pass complexity.

## 5. Experiments

In this section, we empirically validate Anchored Policy Optimization (APO) across three diverse language model architectures and five challenging mathematical reasoning benchmarks. Our experiments are designed to answer two central questions: Can APO effectively mitigate the diversity collapse (RSC) observed in standard RLVR? And does APO's support-constrained rectification enable superior performance trade-offs compared to rigid KL regularization?

### 5.1. Experimental Setup

We conduct our main evaluation using three base models with varying scales and capabilities: **Qwen2.5-7B** (Team, 2024), its mathematics-specialized variant **Qwen2.5-Math-7B** (Yang et al., 2024), and the lightweight **Llama-3.2-3B-Instruct** (AI@Meta, 2024). All models are fine-tuned on the **DAPO-17K** dataset (Yu et al., 2025). We utilize Group Relative Policy Optimization (GRPO) as the underlying policy gradient algorithm. Comprehensive details regarding the training configurations, including batch sizes, learning rates, and APO-specific coefficients, are provided in Appendix D.4. **Baselines.** To rigorously evaluate the effectiveness of our approach, we compare APO against standard **GRPO** and GRPO with KL regularization (**GRPO-KL**). Furthermore, given that our primary motivation focuses on the nuanced handling of negative samples to mitigate the RSC, we introduce **NSR** (Negative Sample Reinforcement) as a specific baseline. Due to the specialized nature of this baseline and resource allocation, we evaluate NSR ex-

*Table 2.* **Main Results on Mathematical Reasoning Benchmarks.** We compare APO against the Base Model, standard GRPO, GRPO with KL regularization (GRPO-KL), and the negative-sample focused baseline NSR (evaluated on Qwen2.5-Math-7B). Performance is reported as **Pass@1** (efficiency, estimated via Avg@K) and **Pass@K** (diversity/coverage), where $K = 256$ for AIME/AMC and $K = 16$ for Math500/Minerva. The **Avg.** column represents the macro-average across all five benchmarks. APO consistently achieves the best balance, improving Pass@1 over the baseline while restoring the Pass@K diversity that standard GRPO sacrifices.

| Model | AIME 24 | | AIME 25 | | AMC 23 | | Math500 | | Minerva | | Avg. | |
|---|---|---|---|---|---|---|---|---|---|---|---|---|
| | Pass@1 | Pass@K | Pass@1 | Pass@K | Pass@1 | Pass@K | Pass@1 | Pass@K | Pass@1 | Pass@K | Pass@1 | Pass@K |
| *Qwen2.5-7B* | | | | | | | | | | | | |
| Base Model | 6.67 | 60.00 | 4.28 | 50.00 | 34.24 | 100.0 | 53.65 | 89.60 | 13.37 | 60.02 | 22.44 | 71.92 |
| GRPO (Baseline) | 14.31 | 50.00 | 12.20 | 50.00 | 67.71 | 97.50 | 78.53 | **91.40** | **40.65** | 63.97 | 42.68 | 70.57 |
| GRPO-KL | 13.72 | 56.67 | 7.37 | **53.33** | 61.26 | **100.0** | 73.83 | 90.00 | 32.97 | 64.71 | 37.83 | 72.94 |
| **APO (Ours)** | **15.85** | **60.00** | **13.50** | 50.00 | **69.75** | **100.0** | **79.60** | 90.00 | 39.17 | **66.91** | **43.57** | **73.38** |
| *Qwen2.5-Math-7B* | | | | | | | | | | | | |
| Base Model | 13.66 | **73.33** | 7.02 | **56.67** | 45.62 | **100.0** | 57.70 | 89.60 | 14.50 | 65.15 | 27.70 | 76.95 |
| GRPO (Baseline) | 32.63 | 70.00 | 14.99 | **56.67** | 69.50 | 97.50 | 81.25 | 93.20 | 45.01 | 65.44 | 48.68 | 76.56 |
| NSR | 32.10 | **73.33** | 14.31 | **56.67** | 70.10 | 97.50 | 81.48 | 93.40 | 44.93 | 66.12 | 48.58 | 77.40 |
| GRPO-KL | 28.59 | 70.00 | 10.68 | 46.67 | 68.04 | **100.0** | 77.68 | 91.40 | 37.68 | **67.28** | 44.53 | 75.07 |
| GRPO-KL (Error-Only) | 29.87 | **73.33** | 12.24 | 43.33 | 75.24 | **100.0** | 79.51 | 92.40 | 37.11 | 66.18 | 46.79 | 75.05 |
| **APO (Ours)** | **33.40** | **73.33** | **16.28** | **56.67** | **75.51** | **100.0** | **82.65** | **94.00** | **45.50** | 66.37 | **50.67** | **78.07** |
| *Llama-3.2-3B-Instruct* | | | | | | | | | | | | |
| Base Model | 6.81 | **36.67** | 0.15 | **30.00** | 24.38 | **85.00** | 44.57 | **75.60** | 17.62 | **48.53** | 18.79 | **55.11** |
| GRPO (Baseline) | 10.85 | 26.67 | 0.20 | 10.00 | 46.00 | 72.50 | 47.91 | 68.00 | **20.43** | 41.91 | 25.08 | 43.82 |
| GRPO-KL | 8.09 | 26.67 | 0.33 | 20.00 | 37.08 | 75.00 | 44.71 | 68.60 | 19.74 | 44.49 | 21.99 | 46.95 |
| **APO (Ours)** | **11.12** | 26.67 | **0.55** | 20.00 | **46.52** | 75.00 | **49.43** | 70.20 | 19.94 | 43.65 | **25.51** | 47.11 |

clusively on the **Qwen2.5-Math-7B** model to analyze its impact on a strong mathematical reasoning backbone.

We evaluate performance on five standard benchmarks: AIME 2024 (Art of Problem Solving, 2025a), AIME 2025 (Art of Problem Solving, 2025a), AMC 2023 (Art of Problem Solving, 2025b), MATH-500 (Hendrycks et al., 2021), and Minerva Math (Lewkowycz et al., 2022). To capture both the accuracy and the exploration capability of the models, we report two metrics: **Pass@1** and **Pass@K**. Specifically, we calculate Pass@1 as the average accuracy across $K$ generated samples (i.e., Avg@K), serving as an unbiased estimator of the policy's expected efficiency. Pass@K denotes the probability of generating at least one correct solution (coverage). We employ a hierarchical evaluation strategy where we use a larger sample budget $K = 256$ for the harder, smaller datasets (AIME 24/25, AMC 23) to ensure statistical significance, while utilizing $K = 16$ for the larger datasets (MATH-500, Minerva Math) where performance estimates stabilize more rapidly.

## 5.2. Main Results and Analysis

Table 2 presents a comparative analysis across all models and benchmarks, empirically delineating the failure path of existing paradigms and the efficacy of APO. The standard GRPO baseline, while effective at boosting sampling efficiency (Pass@1), exhibits a efficiency-diversity trade-off. As visualized in Figure 4, while GRPO improves Pass@1, it

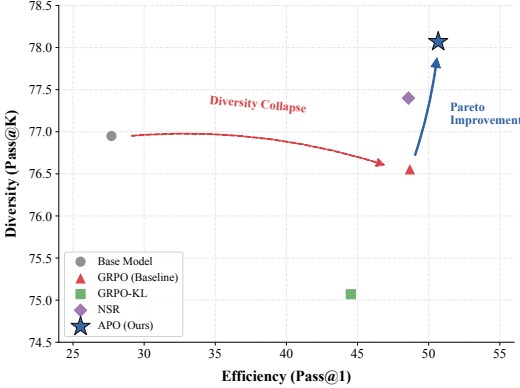

*Figure 4.* **Efficiency-Diversity Pareto Frontier.** Comparison of APO against baselines on Qwen2.5-Math-7B. The dashed red arrow illustrates the *Diversity Collapse* phenomenon observed in standard GRPO, where efficiency gains come at the cost of coverage. The solid blue arrow highlights APO's *Pareto Improvement*, achieving state-of-the-art Pass@1 efficiency while simultaneously restoring the diverse support lost during RL training.

induces a catastrophic collapse in generation diversity. For instance, on Llama-3.2-3B, Pass@K plummets to 43.82%. This sharp inverse correlation confirms that standard negative reinforcement blindly prunes the reasoning tree, sacrificing the model's latent coverage for local sharpness—a hallmark of the *RSC*.

We first examine the **NSR** baseline on Qwen2.5-Math-7B to isolate the impact of negative sample re-weighting. As

shown in Table 2, while NSR marginally mitigates diversity loss (Pass@K: 77.40% vs. 76.56% for GRPO), it fails to translate this exploration into reliable efficiency gains (Pass@1 remains stagnant at 48.58%). This suggests that scalar re-weighting without topological constraints is insufficient to induce "Elastic Reasoning"; the model learns to avoid specific errors but lacks a geometric guide to recover valid alternatives.

**Timing vs. Topology: The Limits of KL Regularization.** To disentangle the impact of *Timing* (regularizing strictly during negative feedback) versus *Topology* (Safe Manifold vs. Shape Matching), we evaluate **GRPO-KL (Error-Only)**. This setup isolates the temporal effect by applying the KL penalty solely to errors, theoretically allowing correct paths to sharpen. This comparison reveals whether the failure of standard regularization stems from its activation schedule or the rigid geometric nature of the KL constraint itself.

The results establish a clear hierarchy of regularization efficacy. While GRPO-KL (Error-Only) outperforms the global GRPO-KL (Pass@1: 46.79% vs. 44.53%), validating that regression should indeed be conditional, a critical anomaly remains: **it still underperforms the unregularized GRPO baseline (48.68%)**.

This finding isolates the fundamental limitation of the KL divergence objective itself. Even when applied conditionally, KL enforces a rigid *Shape Matching* constraint, compelling the policy to mimic the reference model's noise profile rather than simply recovering its valid support. This proves that optimizing the *Timing* of regularization is futile without fundamentally altering the *Topology* of the constraint.

**Breaking the Trade-off with Support Constraints.** APO successfully reconciles this conflict. By combining conditional activation with a support-based constraint, APO achieves a dominant average Pass@1 of **50.67%** on Qwen2.5-Math-7B—significantly outperforming both the efficiency-focused GRPO and diversity-oriented baselines—while maintaining superior coverage (**78.07%** Pass@K). These results confirm that APO does not merely navigate the trade-off but effectively **shifts the Pareto frontier** (Figure 4), enabling aggressive local optimization without the irreversible amputation of valid reasoning branches.

## 6. Analysis

### 6.1. Sensitivity Analysis: Hyperparameters and Manifold Geometry

We validate our geometric hyperparameters on Qwen2.5-Math-7B: anchor set size $K$ (manifold boundary), pull coefficient $\beta$ (restoring force), and push coefficient $\lambda$ (error suppression). We verify our default configuration ($\beta = 0.1, \lambda = 1.05, K = 8$). The overall trends are visualized in Figure 5, and the detailed numerical breakdown is provided in Table 5 of Appendix C.1.

**Geometry of the Safe Manifold ($K$).** $K$ determines the trade-off between strictness and exploration. Consistent with our hypothesis, $K = 8$ provides the optimal trade-off (see Figure 5, Right). Narrower manifolds ($K = 4$) overly restrict valid reasoning paths (Mean Avg 49.46%), while broader manifolds ($K = 16$) introduce tail noise that dilutes the rectification signal, despite minor diversity gains.

**Impact of the Anchor Force ($\beta$).** $\beta$ governs the elasticity of recovery. Comparing our default $\beta = 0.1$ against unanchored ($\beta = 0$) and weaker variants reveals a critical insight: removing the anchor causes a significant performance drop (Mean Avg 48.86%), confirming that blind negative updates are destructive. As shown in Figure 5 (Left), increasing $\beta$ to 0.1 achieves the highest Pass@1 (50.67%), indicating a strong restoring force is necessary to pull trajectories back to the safe manifold. While smaller $\beta$ (e.g., 0.02) yields marginally higher coverage, it fails to correct errors aggressively, reducing overall accuracy.

*Table 3.* **Distributional Health Analysis on Minerva (Llama-3.2-3B-Instruct).** GRPO exhibits *Exploration Collapse*. APO restores entropy and diversity while improving coverage.

| Method | Performance | Output Diversity | | Internal Uncertainty | |
|---|---|---|---|---|---|
| | Pass@K (↑) | Div. Score (↑) | Sem. Sim. (↓) | Entropy (↑) | MaxProb (↓) |
| GRPO | 41.91 | 0.267 | 0.314 | 0.405 | 0.878 |
| **APO (Ours)** | **43.65** | **0.283** | **0.297** | **0.485** | **0.849** |

**Impact of Error Suppression ($\lambda$).** Parameter $\lambda$ controls error suppression intensity. As shown in Figure 5 (Middle) and Table 5, performance follows an inverted U-shape. Standard rejection ($\lambda = 1.0$) yields suboptimal efficiency (49.24%), indicating standard gradients are insufficient for stubborn hallucinations. Conversely, excessive penalty ($\lambda = 1.1$) degrades performance (49.20%) by inducing *Over-pruning* of recoverable paths. $\lambda = 1.05$ offers the precise magnitude to prune incorrect branches without collapsing the manifold.

### 6.2. Micro-Level Analysis: Mitigating Distribution Collapse

We analyze policy behavior on Minerva ($K = 32$) along two axes: Output Diversity and Internal Confidence. We measure diversity via **Diversity Score** ($1 -$ Self-BLEU) (Zhu et al., 2018) and semantic similarity using **Sentence-BERT** (Reimers & Gurevych, 2019). Internal exploration is quantified by **Token Entropy** and **Mean MaxProb**, where high MaxProb signals potential path collapse.

Table 3 highlights the *RSC* in GRPO, evidenced by low entropy (0.405) and high MaxProb (0.878)—signs of over-confidence and path collapse. In contrast, APO restores

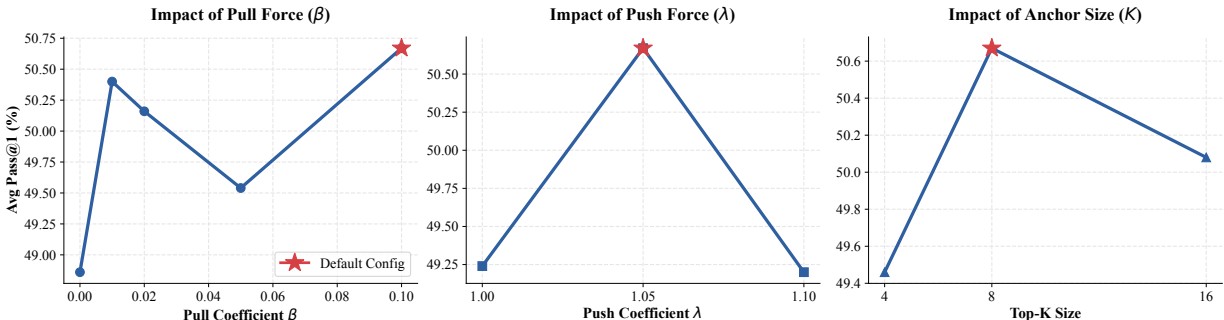

*Figure 5.* **Hyperparameter Sensitivity Analysis.** We visualize the impact of the Pull coefficient $\beta$ (Left), Push coefficient $\lambda$ (Middle), and Anchor Size $K$ (Right) on Pass@1 performance. The Red Star denotes our default configuration. The inverted-U trends in the middle and right plots confirm that APO operates in a optimal trade-off, where the regularization is strong enough to correct errors but compliant enough to permit valid sharpening.

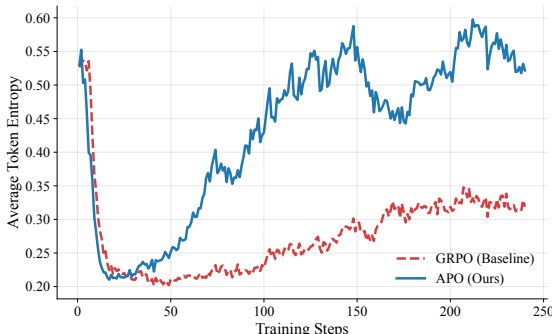

*Figure 6.* **Evolution of Token Entropy.** While both methods experience an initial drop, APO (Blue) demonstrates a robust **entropy recovery** as training progresses, whereas GRPO (Red) remains at a significantly lower entropy level. This confirms that our anchor mechanism effectively revitalizes the exploration potential of the safe manifold.

healthy uncertainty (Entropy 0.485). The temporal evolution of this phenomenon is visualized in Figure 6, showing GRPO's rapid collapse versus APO's sustained exploration. Crucially, this reflects an **Elastic Probability Landscape** rather than confusion: the superior Pass@K (43.65%) confirms that APO retains sufficient probability mass on valid alternative paths, validating that the rectification term effectively prevents irreversible pruning.

# 7. Related Work

## 7.1. Advances in RL and Reasoning for LLMs

Beyond the specific challenge of exploration collapse, the broader integration of Reinforcement Learning into Large Language Models (LLMs) continues to evolve across multiple dimensions. To improve alignment stability and reward accuracy, recent works have proposed adaptive batch-wise scheduling for Direct Preference Optimization (Huang et al., 2025b) and real-time aligned reward frameworks that extend beyond simple semantic matching (Huang et al., 2026b). As models tackle increasingly complex tasks, optimization strategies are adapting to handle extended reasoning chains,

including sequence-level PPO for long-horizon tasks (Wang et al., 2026b) and on-the-fly reasoning paradigms for long-form generation (Wang et al., 2026a). Concurrently, researchers are investigating the intrinsic dynamics of the reasoning process itself, such as understanding optimal implicit stopping conditions (Huang et al., 2026a), while also expanding the RL paradigm into multi-agent collaborative environments (Zhang et al., 2026). Together, these advancements highlight the diverse methodologies driving the current scale and capability of LLM reasoning.

## 7.2. Mitigating Exploration Collapse in RLVR

Output distribution narrowing—often termed "path collapse"—is a primary bottleneck in Reinforcement Learning with Verifiable Rewards (RLVR), severely degrading solution diversity (Pass@k) (Wang et al., 2025b; Yue et al., 2025). Recent literature addresses this through three main strategies.

**Entropy Regularization.** Several works attribute diversity loss to entropy collapse, proposing to explicitly regularize policy entropy to encourage broader exploration (Cui et al., 2025; Cheng et al., 2025; Liang et al., 2025). These methods typically employ auxiliary loss terms to penalize low-entropy distributions, forcing the model to maintain a wider output range.

**Direct Metric Optimization.** Other approaches shift the objective from standard reward maximization to directly optimizing Pass@k (Mahdavi et al., 2025; Walder & Karkhanis, 2025). As Pass@k is a direct proxy for solution diversity, these methods explicitly incentivize the generation of heterogeneous successful trajectories.

**Training Paradigms and Data Augmentation.** A third strategy modifies the macro-level training setup. This includes incorporating diverse datasets (Yan et al., 2025; Dong et al., 2025), tuning hyperparameters for exploration (He et al., 2025; Yu et al., 2025), or interleaving RL with Supervised Fine-Tuning (SFT) to retain base capabilities (Liu

et al., 2025).

### 7.3. Relationship to Our Work

Our proposed Anchored Policy Optimization (APO) offers a perspective **orthogonal** to these strategies. While prior methods explicitly incentivize diversity (via entropy or metric optimization) or augment training data, we address the fundamental mechanics of the policy update relative to the reference model.

We view the decline in Pass@k as a symptom of the *RSC*, where the policy drifts from the reference support. Rather than forcing diversity as an auxiliary objective, APO focuses on **consistency**: anchoring the policy back to the reference model's *Safe Manifold* upon errors. By maximizing Support Coverage, APO recovers diversity naturally as a byproduct of valid exploration and can be seamlessly integrated with the aforementioned methods. A more detailed comparison is provided in Appendix A.1.

## 8. Conclusion

We define RSC as the systemic collapse driven by the interplay of negative squeezing and positive sharpening, identifying it as the root of diversity loss in RLVR. While standard KL fails as a rigid Shape Matching paradigm, our Anchored Policy Optimization (APO) shifts the objective toward Support Coverage within a Safe Manifold. By enabling Elastic Recovery, APO breaks the accuracy-diversity trade-off—maximizing Pass@1 efficiency while revitalizing the Pass@K coverage typically lost to irreversible pruning.

## Impact Statement

This paper presents work whose goal is to advance the field of machine learning. There are many potential societal consequences of our work, none of which we feel must be specifically highlighted here.

## Acknowledgements

This project was supported by National Natural Science Foundation of China (No. 62306132), Guangdong Basic and Applied Basic Research Foundation (No. 2025A1515011564), Natural Science Foundation of Shanghai (No. 25ZR1402136). We thank the anonymous reviewers for their insightful feedback on this work.

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

# A. Extended Discussion and Generalizability

## A.1. Comparison with Existing Mitigation Strategies

In this section, we provide a more granular comparison between Anchored Policy Optimization (APO) and existing mitigation strategies for exploration collapse. We focus specifically on how these methods utilize the reference model and their distinct mechanisms.

**Active Utilization of Reference Knowledge.** The most significant distinction of APO lies in its treatment of the reference model ($\pi_{\text{ref}}$). Standard RL algorithms, such as PPO and GRPO, treat $\pi_{\text{ref}}$ primarily as a passive constraint—a "boundary" defined by KL divergence to prevent excessive drift. In this setting, once the policy collapses into a narrow path (even if that path is suboptimal), the KL penalty merely constrains the policy to remain within the vicinity of that collapsed state, failing to provide a mechanism for recovery. In contrast, APO redefines $\pi_{\text{ref}}$ as an active *anchor* of knowledge. By identifying the *Safe Manifold* within the reference distribution, APO actively utilizes the reference model's logits to construct a "pulling force" when the policy makes errors. This mechanism does not merely penalize deviation; it constructively guides the policy back to valid reasoning paths that may have been forgotten during optimization. This explains the dual improvement observed in our experiments: the "Push" term corrects errors to improve Pass@1, while the "Pull" term restores the rich support of the reference model.

**Orthogonality with Exploration-Centric Paradigms.** Existing literature largely approaches the diversity problem through the lens of *forward exploration*. Methods focusing on Entropy Regularization (Cui et al., 2025) or Direct Metric Optimization (Mahdavi et al., 2025) incentivize the model to widen its output distribution indiscriminately or to directly maximize the probability of diverse successes. Similarly, data-centric approaches focus on expanding the training distribution itself. APO operates on a fundamentally different, orthogonal dimension: the *optimization dynamics*. We do not force the model to explore new, unknown territories blindly; rather, we ensure it retains the known safe regions of the state space. Consequently, APO is compatible with these methods. One could, for instance, employ data augmentation to expand the known manifold and simultaneously use APO to ensure the optimizer covers this expanded manifold effectively without collapsing.

*Table 4.* **Conceptual Comparison of Mitigation Strategies.** We categorize methods based on their primary optimization objective and the role assigned to the reference model. APO is distinct in its use of the reference model as an active guide for manifold restoration rather than a passive constraint or a baseline.

| Method Category | Primary Mechanism | Role of Reference Model |
|---|---|---|
| **Entropy Regularization** | Adds auxiliary loss $\mathcal{H}(\pi)$ to penalize low-entropy distributions and force wider sampling. | **None / Irrelevant**. Diversity is enforced statistically based on the current policy's distribution, ignoring prior knowledge. |
| **Direct Metric Opt.** | Directly optimizes non-differentiable metrics like Pass@k via specialized estimators. | **Baseline**. Typically used implicitly to calculate relative improvements or reward baselines, not for distribution matching. |
| **Data / Training Paradigm** | Augments datasets or interleaves SFT with RL steps to refresh capabilities. | **Prior**. Provides the initial weights but does not participate in the RL update dynamics directly. |
| **APO (Ours)** | Modifies the gradient update via Ratio Rectification (Push-Pull) on error samples. | **Active Anchor**. Explicitly used to map the "Safe Manifold". Its logits provide the target distribution for the recovery (pull) force. |

## A.2. Generalizability to Standard PPO

While our experimental evaluation focuses on Group Relative Policy Optimization (GRPO) due to its state-of-the-art efficiency in the domain of mathematical reasoning (RLVR), it is crucial to note that the theoretical framework of Anchored Policy Optimization (APO) is algorithm-agnostic. The core mechanism of APO—Ratio Rectification—operates exclusively on the policy probability ratio $r_t(\theta) = \pi_\theta(a_t|s_t)/\pi_{\text{old}}(a_t|s_t)$ and the sign of the advantage estimate $A_t$. Consequently, APO is mathematically compatible with any Policy Gradient algorithm that utilizes a trust region or importance sampling, including standard Proximal Policy Optimization (PPO) with a learned value function (Critic). In a standard PPO setting,

the advantage $A_t$ would simply be derived via Generalized Advantage Estimation (GAE) rather than group-relative normalization. We prioritize GRPO in this work solely to align with the current best practices in large-scale reasoning adaptation, where eliminating the Critic model significantly reduces memory overhead and training instability.

### A.3. Discussion on Anchor Validity and Reachability

A potential concern regarding APO is the dependence on the quality of the reference anchor: what if the correct reasoning path lies outside the Top-$K$ Safe Manifold?

We address this through the lens of **Probabilistic Reachability**. In the context of RLVR, the optimization process is primarily a mechanism for extracting and amplifying latent knowledge already present in the Supervised Fine-Tuned (SFT) model, rather than synthesizing new knowledge *ex nihilo*.

If a valid reasoning path is absent from the Top-$K$ support (where $K$ captures the dense probability mass, e.g., $> 95\%$), it typically resides in the heavy tail of the distribution with exponentially decaying probability. Under standard sampling-based exploration, the likelihood of the agent spontaneously discovering such a path is vanishingly small. Therefore, pruning this low-value search space does not theoretically reduce the **effective reachability** of correct solutions. Conversely, by constraining the search to the *Safe Manifold*, APO prevents the policy from drifting into high-entropy hallucination zones, ensuring that the optimization focuses on refining the most plausible reasoning candidates.

## B. Mathematical Foundations of APO

### B.1. Proof of Gradient Alignment And Target Consistency

In this subsection, we provide a rigorous theoretical justification for the Anchored Policy Optimization (APO) objective. We first prove that our heuristic "Pull" term is mathematically equivalent to maximizing support coverage. We then demonstrate via gradient decomposition that the rectified ratio serves as a consistent estimator for a corrected target distribution, and finally analyze the stability of this mechanism under PPO's clipping constraints.

**Proof of Gradient Alignment** We formally derive that the update vector of APO, given a negative advantage, is strictly collinear with the gradient of the Support Coverage objective defined as $\mathcal{J}_{\text{support}}(\theta) = \sum_{y \in \mathcal{M}_{\text{safe}}} \pi_\theta(y)$.

**Proposition B.1** (Gradient Alignment). *In the unclipped regime, the gradient of the APO pull term with respect to a negative advantage is collinear with the gradient of $\mathcal{J}_{support}$.*

*Proof.* Recall that the loss component contributing to the "pull" force is $\mathcal{L}_{\text{pull}} = -\beta A_t \cdot r_{\text{anchor}}$. By definition, $r_{\text{anchor}} \approx \frac{1}{Z_{\text{ref}}} \sum_{k \in S_{\text{anchor}}} \pi_\theta(k)$. Taking the gradient with respect to $\theta$:

$$\nabla_\theta \mathcal{L}_{\text{pull}} = \nabla_\theta \left( -\beta A_t \cdot \frac{1}{Z_{\text{ref}}} \sum_{k \in S_{\text{anchor}}} \pi_\theta(k) \right)$$
$$= \frac{-\beta A_t}{Z_{\text{ref}}} \cdot \nabla_\theta \left( \sum_{k \in S_{\text{anchor}}} \pi_\theta(k) \right). \tag{10}$$

Let $C = \frac{-\beta A_t}{Z_{\text{ref}}}$. Since $\beta > 0$, $Z_{\text{ref}} > 0$, and for negative samples $A_t < 0$, the coefficient $C$ is strictly positive. Thus,

$$\nabla_\theta \mathcal{L}_{\text{pull}} = C \cdot \nabla_\theta \mathcal{J}_{\text{support}}. \tag{11}$$

Consequently, the gradient pushes strictly in the direction of increasing the total probability mass of the Safe Manifold. $\square$

**Theoretical Consistency via Gradient Decomposition** A potential concern is that $\tilde{r}_{\text{APO}}$ is constructed as a linear combination of ratios. Here, we demonstrate that its associated gradient update is theoretically consistent with optimizing a **Rectified Target Distribution**.

Consider the standard policy gradient for a negative sample ($A_t < 0$). Inspired by the gradient decomposition analysis in recent work (Ren & Sutherland, 2025), we interpret the APO update as a first-order approximation of optimizing a mixture

target. Let us define an ideal **Rectified Target Distribution** $Q^*(\cdot|x)$ that reallocates probability mass from the error token $y_{\text{err}}$ to the Safe Manifold:

$$Q^*(y|x) = \begin{cases} 0 & \text{if } y = y_{\text{err}} \\ \frac{\pi_{\text{ref}}(y)}{Z_{\text{safe}}} & \text{if } y \in \mathcal{M}_{\text{safe}} \end{cases} \tag{12}$$

Minimizing the KL divergence between $\pi_\theta$ and $Q^*$ yields a gradient that simultaneously suppresses $y_{\text{err}}$ and increases the mass of $\mathcal{M}_{\text{safe}}$.

Now, observing the gradient of the APO objective in the active region:

$$\nabla_\theta \mathcal{L}_{\text{APO}} \propto A_t \nabla_\theta r_{\text{push}} - \beta A_t \nabla_\theta r_{\text{anchor}}. \tag{13}$$

Substituting the definitions and noting $A_t < 0$:

$$\nabla_\theta \mathcal{L}_{\text{APO}} \propto \underbrace{-|A_t|\nabla_\theta \pi_\theta(y_{\text{err}})}_{\text{Suppress Error}} + \underbrace{\beta|A_t|\nabla_\theta \left( \sum_{k \in \mathcal{M}_{\text{safe}}} \pi_\theta(k) \right)}_{\text{Maximize Safe Mass}}. \tag{14}$$

This decomposition proves that APO's rectification term effectively transforms the dual objectives—Error Suppression ("Push") and Manifold Recovery ("Pull")—into a unified scalar proxy compatible with PPO.

## B.2. Theoretical Derivation of Recursive Space Contraction

In this subsection, we provide a rigorous derivation of the *Recursive Space Contraction* phenomenon. We demonstrate that under the Softmax parameterization, a valid reasoning path with low initial probability enters a mathematical absorbing state in on-policy reinforcement learning. This occurs because the gradient dynamics impose strictly negative updates when competing paths are rewarded, while offering only negligible recovery gradients when competing paths are penalized.

**Preliminaries: Gradient Dynamics of Softmax**    Consider a policy parameterized by logits $z \in \mathbb{R}^{|\mathcal{V}|}$, where $\pi_\theta(y) = \frac{e^{z_y}}{\sum_k e^{z_k}}$. Let $y_t$ denote the token sampled at time $t$, and $A_t$ denote the estimated advantage. The standard policy gradient update for the logit of an arbitrary token $k$ is given by $\Delta z_k \propto A_t \nabla_{z_k} \log \pi_\theta(y_t)$. The gradient of the log-softmax function is:

$$\nabla_{z_k} \log \pi_\theta(y_t) = \delta_{k,y_t} - \pi_\theta(k), \tag{15}$$

where $\delta_{k,y_t}$ is the Kronecker delta. Consequently, the update rule for logit $k$ is:

$$\Delta z_k = \eta A_t(\delta_{k,y_t} - \pi_\theta(k)). \tag{16}$$

We analyze the dynamics of a valid token $y_{\text{valid}}$ that is *not* sampled at the current step (i.e., $y_t \neq y_{\text{valid}}$). Let $\pi_{\text{valid}} \triangleq \pi_\theta(y_{\text{valid}})$. Since $y_t \neq y_{\text{valid}}$, the Kronecker delta $\delta_{y_{\text{valid}},y_t} = 0$, simplifying the update to $\Delta z_{\text{valid}} = -\eta A_t \pi_{\text{valid}}$.

**Proof of Irreversible Collapse**    We now prove that $y_{\text{valid}}$ suffers from irreversible probability decay under both positive and negative feedback loops in an on-policy setting.

**Proposition B.2** (Passive Suppression via Normalization). *Given a sampled token $y_t \neq y_{\text{valid}}$ with positive advantage $A_t > 0$, the probability mass of the unsampled token $y_{\text{valid}}$ strictly decreases, regardless of its validity.*

*Proof.* Substituting $A_t > 0$ into the simplified update rule, we obtain the logit update $\Delta z_{\text{valid}} = -\eta A_t \pi_{\text{valid}}$. Since $\eta, A_t, \pi_{\text{valid}} > 0$, it follows that $\Delta z_{\text{valid}} < 0$. This mathematical property dictates that rewarding any competing token $y_t$ necessitates the suppression of all other tokens to satisfy the normalization constraint $\sum \pi = 1$. Consequently, successful exploration of *other* valid path actively penalizes the latent valid path $y_{\text{valid}}$, driving its logit $z_{\text{valid}}$ to decrease monotonically.    $\square$

The critical failure path arises when the agent samples an error. Standard intuition suggests that penalizing errors should increase the probability of valid alternatives. We prove that this recovery mechanism vanishes for low-probability tokens.

**Proposition B.3** (Vanishing Recovery under Proportional Redistribution). *Given a sampled error token $y_{\text{err}} \neq y_{\text{valid}}$ with negative advantage $A_t = -C$ (where $C > 0$), the recovery gradient for $y_{\text{valid}}$ vanishes as $\pi_{\text{valid}} \to 0$.*

*Proof.* We substitute $A_t = -C$ into the update rule. The update for the valid token becomes $\Delta z_{\text{valid}} = -\eta(-C)\pi_{\text{valid}} = \eta C \pi_{\text{valid}}$. While $\Delta z_{\text{valid}}$ is positive, its magnitude is strictly coupled to the current probability mass $\pi_{\text{valid}}$. We analyze the limit behavior:

$$\lim_{\pi_{\text{valid}} \to 0} \Delta z_{\text{valid}} = \lim_{\pi_{\text{valid}} \to 0} (\eta C \pi_{\text{valid}}) = 0. \tag{17}$$

This linear dependence creates a "Rich-get-Richer" dynamic. The probability mass released by penalizing $y_{\text{err}}$ is redistributed to other tokens proportional to their existing mass. If the policy has already collapsed such that $y_{\text{valid}}$ lies in the tail of the distribution (e.g., $\pi_{\text{valid}} \approx 10^{-5}$), the recovery force $\Delta z_{\text{valid}}$ becomes numerically negligible. In contrast, the dominant path $y_{\text{path}}$ receives an update $\Delta z_{\text{path}} \propto \eta C \pi_{\text{path}}$. Since $\pi_{\text{path}} \gg \pi_{\text{valid}}$, the error penalty paradoxically reinforces the existing path rather than recovering the valid tail. $\qquad\square$

**The Sampling Barrier and Absorbing States** We finally formalize the *Recursive Space Contraction* by integrating the sampling probability. In on-policy RL, the expected gradient update $\mathbb{E}[\Delta z_{\text{valid}}]$ depends on the probability of sampling the trajectory containing $y_{\text{valid}}$.

Let $\mathcal{T}$ be the set of all possible trajectories. The gradient is non-zero only for sampled trajectories. If $\pi_{\text{valid}}$ is suppressed to a threshold $\epsilon$ (via Proposition B.1) such that the expected number of samples $N \cdot \pi_{\text{valid}} < 1$ for a batch size $N$, the valid path effectively ceases to exist in the optimization landscape. Unlike offline settings where the gradient is computed over a fixed dataset $\mathcal{D}$ (guaranteeing $\nabla\mathcal{L}$ exists as long as $(x, y_{\text{valid}}) \in \mathcal{D}$), on-policy RL creates a dependency loop:

$$\pi_{\text{valid}} \to 0 \implies P(\text{Sampling } y_{\text{valid}}) \to 0 \implies \nabla_{\text{recovery}} \to \mathbf{0}. \tag{18}$$

This renders the state $\pi_{\text{valid}} = 0$ an *absorbing state* for the optimization process. Once the policy enters a region where valid paths are unsampled and their logits are suppressed, it cannot escape via standard policy gradients, necessitating the external *Anchor* force introduced in our method.

### B.3. Gradient Dynamics of Probability Redistribution

In this subsection, we provide a rigorous analysis of how probability mass is redistributed under standard Policy Gradient (PG) versus Anchored Policy Optimization (APO). We specifically investigate whether the redistribution mechanism induces a uniform prior or respects the learned topology of the policy, and under what conditions the recovery gradient vanishes.

**The Vanishing Recovery of Standard PG** Consider the standard objective of minimizing the probability of an error token $y_{\text{err}}$, formulated as $\mathcal{L}_{\text{PG}} = \log \pi_\theta(y_{\text{err}})$. The gradient with respect to the logit $z_k$ of any alternative token $k$ ($k \neq y_{\text{err}}$) is given by:

$$\nabla_{z_k} \mathcal{L}_{\text{PG}} = -\pi_\theta(y_{\text{err}}) \cdot \pi_\theta(k). \tag{19}$$

This update rule exposes two fundamental limitations in the context of error correction. **First**, the update magnitude exhibits a *Proportional Dependence*, often described as a "Rich-get-Richer" dynamic. Since the gradient for token $k$ is directly proportional to its current probability $\pi_\theta(k)$, if the policy has already collapsed to a local path where valid alternatives have negligible support ($\pi_\theta(k) \approx 0$), the recovery gradient effectively vanishes. This mathematical property confirms the *Squeezing Effect*: the model becomes incapable of "seeing" or recovering valid paths once they are sufficiently suppressed. **Second**, the driving force of the update is scaled by the error probability $\pi_\theta(y_{\text{err}})$. As the optimization successfully suppresses the error (i.e., $\pi_\theta(y_{\text{err}}) \to 0$), this redistribution mechanism prematurely terminates. Consequently, the probability mass is not necessarily reallocated to valid alternatives, but rather remains trapped in the local path, as the gradient signal disappears before exploration can occur.

**The Active Inflation of APO** In contrast, APO introduces the anchor maximization term $\mathcal{J}_{\text{support}} = \sum_{j \in S_{\text{anchor}}} \pi_\theta(j)$. Let $P_{\text{safe}}$ denote the total probability mass accumulated within the safe manifold. The gradient with respect to the logit $z_k$ of a specific valid token $k \in S_{\text{anchor}}$ is derived as:

$$\begin{aligned}
\nabla_{z_k} \mathcal{J}_{\text{support}} &= \nabla_{z_k} \left( \sum_{j \in S_{\text{anchor}}} \frac{e^{z_j}}{\sum_i e^{z_i}} \right) \\
&= \pi_\theta(k) - \pi_\theta(k) \sum_{j \in S_{\text{anchor}}} \pi_\theta(j) \\
&= \pi_\theta(k) \cdot (1 - P_{\text{safe}}).
\end{aligned} \tag{20}$$

This derivation highlights the superior redistribution mechanics of APO. Crucially, the mechanism performs **Structure-Preserving Inflation**. Similar to standard PG, the update remains proportional to $\pi_\theta(k)$, ensuring that the policy's internal ranking of tokens is respected. APO does not flatten the distribution into a uniform prior—which would destroy learned knowledge—but rather "inflates" the probability of the entire valid structure proportionally to its existing shape.

Furthermore, APO resolves the vanishing gradient problem through a **Persistent Recovery Signal**. The driving force of the update is scaled by the term $(1 - P_{\text{safe}})$, which represents the probability mass leaked outside the safe manifold. Unlike standard PG, which halts once the specific error is suppressed, APO maintains a strong restorative gradient as long as the total mass of the safe manifold is not maximized ($P_{\text{safe}} < 1$). This guarantees that the pruning of errors is inextricably linked to the robust recovery of the reference model's support, preventing the irreversible collapse observed in baselines.

### B.4. Stability Analysis: KL Penalty vs. APO Ratio Rectification

To formally address why standard KL regularization can destabilize the policy by deviating from the **Trust Region**, we compare the gradient update vectors of KL-regularized PG and APO.

**Case 1: KL-regularized Policy Gradient.** In standard implementations, the KL divergence is added as an auxiliary penalty term:

$$\mathcal{L}_{\text{total}}(\theta) = \mathcal{L}_{\text{CLIP}}(\theta) - \eta D_{\text{KL}}(\pi_\theta \| \pi_{\text{ref}}). \tag{21}$$

The resulting gradient is a linear combination of two potentially opposing vectors:

$$\nabla_\theta \mathcal{L}_{\text{total}} = \nabla_\theta \mathcal{L}_{\text{CLIP}} + \eta \nabla_\theta \mathcal{L}_{\text{KL}}. \tag{22}$$

As illustrated in Figure 2(a), even if $\nabla_\theta \mathcal{L}_{\text{CLIP}}$ is zero due to the ratio hitting the clipping boundary $1 \pm \epsilon$, the KL gradient $\nabla_\theta \mathcal{L}_{\text{KL}}$ remains non-zero. This creates a **Gradient Conflict**: the KL force continues to push the parameters $\theta$ without being constrained by the PPO trust region. Consequently, the resultant update can move the importance sampling ratio $r_t(\theta)$ significantly beyond the $[1 - \epsilon, 1 + \epsilon]$ window, leading to high-variance updates and training instability.

**Case 2: APO Ratio Rectification.** In contrast, APO embeds the restoring force directly into the policy ratio $\tilde{r}_{\text{APO}}$ (Eq. 9). The objective remains a single clipped surrogate:

$$\mathcal{L}_{\text{APO}}(\theta) = \mathbb{E}\left[\min\left(\tilde{r}_{\text{APO}}(\theta) A_t, \text{clip}(\tilde{r}_{\text{APO}}(\theta), 1 - \epsilon, 1 + \epsilon) A_t\right)\right]. \tag{23}$$

By the chain rule, the gradient is:

$$\nabla_\theta \mathcal{L}_{\text{APO}} = \begin{cases} A_t \nabla_\theta \tilde{r}_{\text{APO}}(\theta) & \text{if } \tilde{r}_{\text{APO}} \in [1 - \epsilon, 1 + \epsilon] \\ 0 & \text{otherwise} \end{cases} \tag{24}$$

**Mathematical Guarantee of Trust Region Adherence:** Unlike the KL penalty, the APO "Pull Force" is *intrinsically capped*. If the rectification term attempts to pull the policy further than the trust region allows, the objective hits the clipping flat-top, and the gradient becomes exactly zero. This ensures that the recovery process is **Self-Stabilizing**: the policy is restored to the safe manifold at the maximum rate permitted by the trust region, but it is mathematically impossible for the auxiliary anchor force to push the update into an unstable regime. This provides a formal justification for the improved training stability observed in our experiments.

**Analysis of the Clipped Region.** If the rectification force is strong enough such that $\tilde{r}_{\text{APO}} < 1 - \epsilon$, the objective becomes constant $((1 - \epsilon) A_t)$ and gradients vanish. While superficially resembling gradient vanishing, within the PPO framework, the clipping bound $1 - \epsilon$ represents the **maximum safe penalty** permissible in a single update step.

Therefore, if APO causes the ratio to hit this lower bound, it indicates that the "Pull Force" has successfully amplified the error signal to the maximum robust magnitude allowed by the trust region. In this sense, the zero gradient confirms that the policy has been *sufficiently penalized*, preventing the restoring force from destabilizing the policy updates via excessively large steps.

# C. Extended Empirical Results

## C.1. Sensitivity Analysis: $K, \beta, \lambda$

In this section, we provide the comprehensive numerical data corresponding to the sensitivity analysis discussed in Section 6.1. Table 5 details the Pass@1 and Pass@K performance across all five benchmarks for variations in the Anchor Size ($K$), Pull Coefficient ($\beta$), and Push Coefficient ($\lambda$).

*Table 5.* **Hyperparameter Sensitivity Analysis on Qwen2.5-Math-7B.** Detailed numerical breakdown of the impact of Pull Coefficient ($\beta$), Push Coefficient ($\lambda$), and Anchor Size ($K$). The **Default** configuration ($K = 8, \beta = 0.1, \lambda = 1.05$) yields the highest Mean Average accuracy (Pass@1). Bold values indicate the best performance in each column.

| Config | AIME24 | AIME25 | AMC23 | Math500 | Minerva | Average | |
| --- | --- | --- | --- | --- | --- | --- | --- |
| (Params) | M / P | M / P | M / P | M / P | M / P | (Pass@1) | (Pass@K) |
| **Default (Ours)** | **33.4** / 73.3 | 16.3 / 56.7 | 75.5 / **100** | 82.7 / **94.0** | 45.5 / 66.4 | **50.67** | 78.07 |
| $K = 8, \beta = 0.1, \lambda = 1.05$ | | | | | | | |
| *Varying Pull Coefficient ($\beta$) [Fix $\lambda = 1.05, K = 8$]* | | | | | | | |
| $\beta = 0$ (No Anchor) | 28.0 / 70.0 | 15.6 / 53.3 | 73.1 / **100** | 82.9 / 92.6 | 44.7 / 65.4 | 48.86 | 76.26 |
| $\beta = 0.01$ | 31.5 / 73.3 | 15.3 / 53.3 | 75.1 / **100** | 83.3 / 93.4 | 46.8 / 66.9 | 50.40 | 77.38 |
| $\beta = 0.02$ | 31.3 / **80.0** | 16.3 / 56.7 | **76.3** / 97.5 | 81.8 / 92.8 | 45.1 / 65.8 | 50.16 | 78.56 |
| $\beta = 0.05$ | 31.0 / 73.3 | 15.8 / **60.0** | 74.4 / 97.5 | 82.0 / 92.8 | 44.5 / **68.4** | 49.54 | 78.40 |
| *Varying Push Coefficient ($\lambda$) [Fix $\beta = 0.1, K = 8$]* | | | | | | | |
| $\lambda = 1.0$ (Std.) | 28.1 / 70.0 | 16.6 / 56.7 | 72.2 / 97.5 | **83.5** / 93.4 | 45.8 / 66.9 | 49.24 | 76.90 |
| $\lambda = 1.1$ (Strong) | 29.3 / **80.0** | **17.2** / **60.0** | 72.3 / 97.5 | 82.2 / 92.8 | 45.0 / 67.3 | 49.20 | **79.52** |
| *Varying Anchor Size ($K$) [Fix $\beta = 0.1, \lambda = 1.05$]* | | | | | | | |
| $K = 4$ | 29.9 / 70.0 | 15.7 / 53.3 | 73.2 / 97.5 | 82.7 / 93.4 | 45.8 / 66.2 | 49.46 | 76.08 |
| $K = 16$ | 29.7 / 70.0 | 15.9 / 56.7 | 74.7 / **100** | 83.1 / 93.4 | **47.0** / 66.9 | 50.08 | 77.40 |

# D. Implementation Details and Exclusive Anchoring

## D.1. Analysis of Signal Cancellation

In Section 3.3, we introduced the **Exclusive Anchoring** mechanism, which excludes the current error token $y_t$ from the anchor set $S_{\text{anchor}}$. Here, we mathematically demonstrate why this exclusion is necessary to prevent gradient cancellation.

Consider a "Naive Anchor" setup where the error token is included: $S_{\text{naive}} = S_{\text{anchor}} \cup \{y_t\}$. The total gradient update for the logit $z_{y_t}$ of the error token would be a summation of the standard Policy Gradient (Push) and the Anchor Gradient (Pull).

For a negative advantage $A_t < 0$, the standard PG objective seeks to decrease $\pi_\theta(y_t)$:

$$\nabla_{z_{y_t}} \mathcal{L}_{\text{PG}} \approx A_t \cdot \pi_\theta(y_t)(1 - \pi_\theta(y_t)) < 0 \quad \text{(Since } A_t < 0\text{)}. \tag{25}$$

However, the Anchor term seeks to maximize the sum of probabilities in $S_{\text{naive}}$. If $y_t \in S_{\text{naive}}$, the gradient contribution from the anchor term with respect to $z_{y_t}$ is positive:

$$\nabla_{z_{y_t}} \mathcal{L}_{\text{anchor}} \propto -\beta A_t \cdot \nabla \pi_\theta(y_t) > 0 \quad \text{(Since } -\beta A_t > 0\text{)}. \tag{26}$$

Thus, the net gradient on the error token becomes:

$$\nabla_{\text{net}} \approx \underbrace{-|A_t| \cdot \text{Grad}_{\text{push}}}_{\text{Suppress}} + \underbrace{\beta|A_t| \cdot \text{Grad}_{\text{pull}}}_{\text{Boost}} . \tag{27}$$

This creates a direct **Signal Cancellation**: the anchor term actively fights against the suppression of the error. In extreme cases (e.g., strong $\beta$), this can neutralize the error signal entirely or even encourage the model to repeat the error to satisfy the anchor constraint.

By enforcing $S_{\text{anchor}} = \text{TopK} \setminus \{y_t\}$, we ensure that $\nabla_{z_{y_t}} \mathcal{L}_{\text{anchor}}$ is strictly negative (via the softmax normalization term $\partial \pi_k / \partial z_{y_t} = -\pi_k \pi_{y_t}$), thereby aligning the "Push" and "Pull" forces to consistently suppress the error while boosting *alternative* candidates.

### D.2. Expected Sampling Dynamics and Manifold Purification

A potential theoretical concern regarding Anchored Policy Optimization (APO) is whether the restorative "Pull" force inadvertently boosts the probability of latent error tokens that happen to reside within the Safe Manifold ($\mathcal{M}_{\text{safe}}$), thereby counteracting the pruning process. To address this, we provide a formal gradient analysis from the perspective of expected sampling updates. We demonstrate that the expected suppressive force mathematically dominates the collateral boost, ensuring the manifold is effectively purified over time.

**Notation and Preliminaries**     Let us formalize the optimization state at a given reasoning step $s$. We analyze the gradient dynamics with respect to the logit $z_e$ of a specific latent error token $e$ residing in the Top-$K$ manifold ($e \in \mathcal{M}_{\text{safe}}$). Under the current policy distribution $\pi_\theta(\cdot|s)$, let $y_t \sim \pi_\theta(\cdot|s)$ denote the token sampled at step $t$. We define $P_{\text{corr}}$ as the total probability mass of all valid or correct tokens, and $P_{\text{err}\setminus\{e\}}$ as the total probability mass of all *other* error tokens, such that the remaining probability mass satisfies $1 - \pi_\theta(e|s) = P_{\text{corr}} + P_{\text{err}\setminus\{e\}}$. Finally, let $A_t$ represent the estimated advantage; for analytical simplicity, we assume errors receive a constant negative advantage $A_t = -C$ where $C > 0$.

**The Expected Pull Force**     When the policy samples *another* error token $y_t = y' \neq e$, two mechanisms lift the probability of $e$. First, penalizing $y'$ naturally redistributes mass to all other tokens via standard Softmax redistribution. Second, because $e$ is in the anchor set and $y' \neq e$, Exclusive Anchoring triggers a restorative APO Pull Force on $e$. Following standard policy gradient dynamics and the gradient of our Support Coverage objective (Equation 20), the gradient on $z_e$ when $y'$ is sampled is:

$$\Delta z_e(y') = C\pi_\theta(e|s) + \frac{\beta C}{Z_{\text{ref}}}\pi_\theta(e|s)(1 - P_{\text{safe}}) \tag{28}$$

Taking the expectation over all other error samples $y' \in \text{Errors} \setminus \{e\}$, the total expected lift-up force is:

$$\mathbb{E}_{\text{lift}}[\Delta z_e] = P_{\text{err}\setminus\{e\}} \cdot \left[ C\pi_\theta(e|s)\left(1 + \frac{\beta}{Z_{\text{ref}}}(1 - P_{\text{safe}})\right) \right] \tag{29}$$

**The Expected Push Force**     Because the probability of token $e$ is lifted, it becomes more likely to be sampled. When the policy explicitly samples $y_t = e$, it receives the negative advantage $A_t = -C$, and standard PG strongly suppresses it. (Note: Exclusive Anchoring strictly excludes $e$ from the anchor set in this scenario, meaning the APO pull term does not boost $e$, and in fact slightly suppresses it. To establish a strict lower bound for suppression, we conservatively consider only the PG push term).

The probability of sampling $e$ is $\pi_\theta(e|s)$, and the PG gradient on $z_e$ is $-C(1 - \pi_\theta(e|s))$. Therefore, the expected push-back force is:

$$\mathbb{E}_{\text{push}}[\Delta z_e] = \pi_\theta(e|s) \cdot [-C(1 - \pi_\theta(e|s))] \tag{30}$$

**Net Dominance Bound**     For Exclusive Anchoring to effectively purify the manifold, the expected "Push Back" magnitude must strictly dominate the expected "Lift Up" magnitude:

$$|\mathbb{E}_{\text{push}}[\Delta z_e]| > |\mathbb{E}_{\text{lift}}[\Delta z_e]| \tag{31}$$

Substituting our derivations yields:

$$C\pi_\theta(e|s)(1 - \pi_\theta(e|s)) > C\pi_\theta(e|s)P_{\text{err}\setminus\{e\}}\left(1 + \frac{\beta}{Z_{\text{ref}}}(1 - P_{\text{safe}})\right) \tag{32}$$

Dividing both sides by $C\pi_\theta(e|s)$:

$$1 - \pi_\theta(e|s) > P_{\text{err}\setminus\{e\}} + P_{\text{err}\setminus\{e\}}\frac{\beta}{Z_{\text{ref}}}(1 - P_{\text{safe}}) \tag{33}$$

By definition, $1 - \pi_\theta(e|s) = P_{\text{corr}} + P_{\text{err}\setminus\{e\}}$. Substituting this identity into the left side:

$$P_{\text{corr}} + P_{\text{err}\setminus\{e\}} > P_{\text{err}\setminus\{e\}} + P_{\text{err}\setminus\{e\}} \frac{\beta}{Z_{\text{ref}}}(1 - P_{\text{safe}}) \tag{34}$$

Canceling $P_{\text{err}\setminus\{e\}}$ from both sides establishes the final condition for manifold purification:

$$P_{\text{corr}} > P_{\text{err}\setminus\{e\}} \frac{\beta}{Z_{\text{ref}}}(1 - P_{\text{safe}}) \tag{35}$$

**Conclusion.** The left side of the inequality represents the model's base capability on valid paths, while the right side quantifies the APO collateral boost factor. Under our default hyperparameters ($K = 8, \beta = 0.1$) and typical reference confidence ($Z_{\text{ref}} \approx 0.95$) with $P_{\text{safe}} > 0$, the multiplying factor $\frac{\beta}{Z_{\text{ref}}}(1 - P_{\text{safe}})$ is strictly bounded below roughly 0.11.

Consequently, for the restorative push-back force to mathematically overpower the collateral lift-up force, the policy only needs to maintain a valid probability mass ($P_{\text{corr}}$) greater than approximately 10% of the *other* error mass ($P_{\text{err}\setminus\{e\}}$). Because this represents an exceptionally low threshold, the iterative sampling process inherently purifies the anchor set of latent errors without inducing irreversible collapse.

### D.3. Computational Cost Analysis

To address the practical overhead of our proposed method, we provide a comparison of the computational costs associated with each training algorithm. Table 6 details the total GPU hours and relative cost for training the models.

Standard Kullback-Leibler (KL) regularization requires dense gradient propagation over the full vocabulary dimension (e.g., $V \approx 128k$), which introduces a noticeable computational burden. In contrast, Anchored Policy Optimization (APO) operates exclusively on a highly sparse manifold defined by the anchor set (e.g., $K = 8$). Because we only calculate gradients for these sparse Top-$K$ tokens rather than the full vocabulary, APO introduces virtually no additional overhead compared to standard KL-regularized baselines, while significantly improving both training stability and inference performance.

*Table 6.* **Computational Cost Comparison.** Total GPU hours and relative training costs. Despite employing a restorative anchor mechanism, APO's sparse manifold constraint ensures its computational cost remains slightly lower than standard KL regularization.

| Training Algorithm | Total GPU Hours | Relative Cost vs. Baseline |
|---|---|---|
| GRPO (Baseline) | 96 | 1.00× |
| GRPO-KL | 115 | 1.20× |
| **APO (Ours)** | 114 | 1.19× |

### D.4. Experimental Details

We provide the detailed hyperparameters used for the reinforcement learning stage in Table 7. **Unless otherwise specified, all experiments and algorithms reported in this paper utilize this unified configuration to ensure fair comparison.**

Specifically, we implement Group Relative Policy Optimization (GRPO) with a standard clipping mechanism, setting the clip ratio $\epsilon = 0.2$. Furthermore, the policy loss is computed using `token_mean` aggregation (averaging loss over valid tokens) rather than sample summation, to stabilize training variance across variable-length reasoning paths. Our code is implemented based on the `verl` (Sheng et al., 2025) repository.

*Table 7.* **Hyperparameter Configuration.** Detailed training parameters and coefficients used in our experiments.

| Hyperparameter | Value |
| --- | ---: |
| ***Training Configuration*** | |
| Global Batch Size | 512 |
| Mini-Batch Size | 32 |
| Learning Rate | 1e-6 |
| Group Size ($N$) | 8 |
| Clip Ratio ($\epsilon$) | 0.2 |
| Loss Aggregation | Token Mean |
| Max Context Length | 2048 |
| Max Prompt Length | 512 |
| Total Training Steps | 240 |
| Training Platform | $32 \times$ NVIDIA A800 80G |
| ***Method Coefficients*** | |
| APO Push Coefficient ($\lambda$) | 1.05 |
| APO Pull Coefficient ($\beta$) | 0.1 |
| APO Anchor Size ($K$) | 8 |
| KL Coefficient (Standard) | 0.01 |
| KL Coefficient (Error-Only) | 0.01 |

