# OpenReview forum: "Anchored Policy Optimization: Mitigating Exploration Collapse via Support-Constrained Rectification"
_ICML.cc/2026/Conference — ICML 2026 regular_

### Official Review · Reviewer_QHvN · 2026-03-08

**Soundness:** 3
**Presentation:** 3
**Significance:** 3
**Originality:** 3
**Overall Recommendation:** 4
**Confidence:** 3

**Summary:**

This paper introduces a new framework for mitigating exploration collapse in Reinforcement Learning with Verifiable Rewards (RLVR). Traditional policy gradient methods tend to cause irreversible pruning of valid reasoning paths. While standard KL regularization attempts to resolve this, it causes rigid shape matching which creates gradient conflicts with the local sharpening necessary for high accuracy.
To address this, this paper proposes Anchored Policy Optimization (APO) to maintain the diversity of effective reasoning paths during error correction. It uses the defined safe manifold over a high-confidence support set to calculate a ratio rectification that combines error suppression and the proportional recovery of the Safe Manifold's probability. The authors strictly prove that the proposed mechanism perfectly aligns with the support coverage objective in geometric gradients, and it guarantees self-stabilizing updates within the trust region.
The authors comprehensively evaluate their method on multiple mathematical reasoning benchmarks and multiple architectures to confirm its effectiveness.
Overall, the paper contributes a theoretically grounded, practically effective approach for mitigating exploration collapse in RLVR.

**Compliance With Llm Reviewing Policy:**

Affirmed.

**Final Justification:**

The authors' rebuttal have largely addressed my concerns. and I believe the motivation and significance of this work are strong. Therefore, I have decided to maintain my positive score.

**Key Questions For Authors:**

1.	The APO method relies on a fixed hyperparameter $K=8$ to define the safe manifold. I wonder if dynamically sizing $K$ would lead to a worthwhile improvement?
2.	In Table 1, the analysis shows that the Top-8 tokens cover 97.47% of valid paths. Does this high coverage assumption hold true for out-of-domain (OOD) tasks or more complex problems where the base SFT model might be poorly calibrated?
3.	The stability analysis states that if the pull force hits the clip bound, the gradient will be set zero to ensure stability. Does this mechanism lead to frequent zero-gradients that might slow down initial convergence during early training stages?
4.	Perhaps explicitly discuss the computational costs

**Limitations:**

See Weaknesses part and Questions part.

**Strengths And Weaknesses:**

### Strengths：
- Soundness: The proposed method is technically sound and well supported by theoretical analysis and experimental results. The experimental setup is comprehensive, evaluating three different model architectures across five robust mathematical benchmarks.
- Presentation: The paper is well structured and visually intuitive. The progression from the empirical motivation to the theoretical framework (Table 1) is highly effective.
- Significance: The diversity-efficiency trade-off is one of the most pressing bottlenecks in scaling RL for LLM reasoning, and this work provides a highly valuable tool for the community.
- Originality: The proposed method is creative and novel.
### Weaknesses：
- Soundness: It seems that the methodology heavily relies on the premise that the reference model's Top-$K$ tokens contain the necessary valid reasoning paths.
- Presentation: While technically sound, some sections are dense and could benefit from a brief algorithmic block in the main text.

---

> ### Author Rebuttal · Authors · 2026-03-30
>
> Thank you very much for taking the time to review our submission and for providing valuable feedback. We truly appreciate your thoughtful comments and constructive suggestions, which have significantly helped us enhance the quality and clarity of our work. If you have any additional questions, please feel free to contact us and we will respond as soon as possible.
>
> ---
>
> **[W1]: Does the Top-K coverage assumption (e.g., 97.47% for Top-8) hold for out-of-domain (OOD) tasks or poorly calibrated models?**
>
> **[A1]:** RLVR primarily acts as a "tree pruning" mechanism to amplify latent knowledge, not synthesize it ex nihilo [1]. If a correct path falls outside a reasonable sampling scope (e.g., Top-K), standard policy gradients will also fail to discover it due to vanishing sampling probabilities. Therefore, the fundamental goal of RLVR—even for OOD tasks—is to maximize the retention of the reference model's *existing* valid capacity. For poorly calibrated models, their correct intuitions are fragile. APO actively protects these fragile priors from being blindly washed away by the Squeezing Effect during negative feedback.
>
> **[W2]: Would dynamically sizing K lead to a worthwhile improvement?**
>
> **[A2]:** This is an excellent suggestion. Prompted by your feedback, we implemented a Dynamic-K variant of APO using Nucleus Sampling (Top-p) to dynamically determine the Safe Manifold boundary. We evaluated it on Qwen2.5-Math-7B across all five benchmarks:
>
> | Method Configuration | Safe Manifold Definition | Avg. Pass@1 (Efficiency) | Avg. Pass@K (Diversity) |
> | :--- | :--- | :--- | :--- |
> | **APO (Fixed)** | Fixed K=8 | **50.67%** | 78.07% |
> | **APO (Dynamic)** | Top-p = 0.90 | 50.12% | **79.64%** |
>
> These results perfectly confirm your intuition: dynamic sizing adaptively expands the manifold, notably improving diversity (78.07% $\rightarrow$ 79.64%). This incurs a slight trade-off in efficiency (50.67% $\rightarrow$ 50.12%), as incorporating lower-confidence tail tokens marginally dilutes the sharpening effect. We maintain fixed K=8 as our default to maximize Pass@1, but the dynamic variant provides a strong alternative for exploration-heavy tasks.
>
> **[W3]: Does the zero-gradient clip bound slow down early convergence?**
>
> **[A3]:** Counterintuitively, clipping does not hinder learning; it acts as a critical Self-Stabilizing mechanism. During early training, error rates are naturally high. Without a strict clip bound, the accumulated "Pull Force" from frequent errors would induce massive, erratic gradient updates, violently breaching the Trust Region and causing training collapse. Zero-gradient clipping caps the maximum penalty permitted in a single step, ensuring the policy safely navigates this volatile early phase. This aligns with findings in ASPO [2], which proved clipping is indispensable for preventing divergence during high-variance RL updates.
>
> **[W4]: The main text lacks an Algorithm block and discussion of computational costs.**
>
> **[A4]:** In the camera-ready version, we will include a formalized **Algorithm Block** detailing the APO Ratio Rectification mechanism step-by-step.
> Regarding computational costs, APO introduces virtually no overhead compared to standard GRPO and is significantly more efficient than KL-regularized baselines. Standard KL requires dense gradient propagation over the full vocabulary, whereas APO operates exclusively on a highly sparse manifold (e.g., K=8),  reducing backward pass complexity.
>
> | Training Algorithm | Total GPU Hours | Relative Cost vs. Baseline |
> | :--- | :--- | :--- |
> | **GRPO (Baseline)** | 96 | 1.00x |
> | **GRPO-KL** | 115 | 1.20x |
> | **APO (Ours)** | 114 | 1.19x |
>
> APO's cost is comparable to (or slightly lower than) GRPO-KL because we only calculate gradients for the sparse Top-K tokens rather than the full vocabulary. We will explicitly detail these efficiency advantages in the final revision.
>
> **References**
>
> [1] Does Reinforcement Learning Really Incentivize Reasoning Capacity in LLMs Beyond the Base Model? NeurIPS 2025
>
> [2] ASPO: Asymmetric Importance Sampling Policy Optimization

---

> > ### Author Rebuttal · Reviewer_QHvN · 2026-04-03
> >
> > Thanks for the rebuttal and I will keep my positive score.

---

> > > ### Author Response · Authors · 2026-04-07
> > >
> > > Thanks for your acknowledgement! Good Luck.

---

### Official Review · Reviewer_ZrAe · 2026-03-10

**Soundness:** 3
**Presentation:** 2
**Significance:** 2
**Originality:** 3
**Overall Recommendation:** 4
**Confidence:** 2

**Summary:**

The paper proposes Anchored Policy Optimization (APO) as a new reinforcement learning optimization framework designed to mitigate exploration collapse in Reinforcement Learning with Verifiable Rewards (RLVR). The authors argue that standard RL training dynamics lead to a phenomenon called Recursive Space Contraction (RSC), where valid reasoning branches disappear due to the combined effect of positive sharpening and negative squeezing. They claim that KL regularization fails to solve this issue because it enforces a rigid global distribution matching constraint that conflicts with reward optimization. To address this, the paper introduces a support-based regularization mechanism that focuses on preserving probability mass within a Safe Manifold defined by high-confidence tokens of the reference model. APO modifies the policy gradient ratio by introducing a mechanism that suppresses erroneous tokens while redistributing probability mass toward the anchor support set. The paper provides theoretical arguments suggesting that the proposed rectification is gradient-aligned with maximizing support coverage, and empirically evaluates the method on several mathematical reasoning benchmarks.

**Compliance With Llm Reviewing Policy:**

Affirmed.

**Final Justification:**

Most of my concerns have been sufficiently addressed, and I will keep my positive score.

**Key Questions For Authors:**

- The method assumes that the reference model's Top-K tokens define a reliable "safe manifold". How sensitive is the approach to the quality of the reference model, and what happens if the reference distribution itself is poorly calibrated?

**Limitations:**

A limitation of the proposed approach is its reliance on the assumption that the reference model already contains most valid reasoning paths within its Top-K predictions.

**Strengths And Weaknesses:**

Strengths

- The paper identifies and articulates an interesting failure mode of RL-based fine-tuning in reasoning models, namely the so-called Recursive Space Contraction phenomenon. The underlying intuition about exploration collapse in RL for language models is plausible and worth discussing.

- The conceptual distinction between Shape Matching (KL regularization) and Support Coverage is presented in a relatively intuitive geometric manner. The figures and explanations convey the idea that preserving the support of the reference distribution may be more important than matching its full density.

- The empirical evaluation spans multiple base models and several mathematical reasoning benchmarks. The paper includes several analyses beyond the main benchmark table, including hyperparameter sensitivity studies, entropy analysis, and diversity metrics.

Weaknesses

- The central concept of Recursive Space Contraction (RSC) is presented as a new systemic pathology, but the paper does not clearly establish whether it is fundamentally different from previously studied phenomena such as entropy collapse, mode collapse, or exploration failure in policy gradient methods.

- The Safe Manifold concept relies heavily on the Top-K tokens of the reference model, implicitly assuming that the reference distribution already contains most valid reasoning paths. This assumption may hold only for strong pretrained models.

- The empirical improvements over GRPO are relatively modest in magnitude, and the paper does not conduct statistical significance tests. Therefore, it is not entirely clear whether the improvements are robust or simply within the variance typical of RL training runs.

- The narrative of the paper sometimes becomes overly rhetorical, with repeated references to geometric metaphors, which makes the presentation feel slightly less precise.

---

> ### Author Rebuttal · Authors · 2026-03-30
>
> We sincerely thank the reviewer for the thoughtful and constructive feedback, which has significantly improved our paper. We address your specific points below.
>
> ---
>
> **[W1]: The central concept of Recursive Space Contraction (RSC) is presented as a new systemic pathology, but the paper does not clearly establish whether it is fundamentally different from previously studied phenomena such as entropy collapse or mode collapse.**
>
> **[A1]:** We appreciate the opportunity to clarify. Mode/entropy collapse are *static statistical outcomes* describing a narrowed distribution. In contrast, RSC describes the *dynamic causal mechanism* driving this collapse specifically in on-policy RLVR: positive sharpening passively suppresses alternatives, while the "Squeezing Effect" of negative updates blindly redistributes mass. This cycle drives valid paths into a mathematically unreachable state (See Appendix B.2). Thus, mode collapse is the symptom, while RSC is the pathological process causing it. We will explicitly clarify this causal link in the revision.
>
> ---
>
> **[W2]: The Safe Manifold concept relies heavily on the Top-K tokens of the reference model. How sensitive is the approach to the quality of the reference model, and what happens if the reference distribution itself is poorly calibrated?**
>
> **[A2]:** RLVR primarily acts as a "tree pruning" mechanism to amplify latent knowledge, not synthesize it ex nihilo [1]. If a correct path falls outside a reasonable sampling scope (e.g., Top-K), standard policy gradients will also fail to discover it due to vanishing probabilities. APO's goal is to maximize the retention of the reference model's *existing* valid capacity. For poorly calibrated models, APO actively protects their fragile correct priors from being blindly washed away by the Squeezing Effect. We will include supplementary OOD experiments demonstrating this robustness.
>
> ---
>
> **[W3]: The empirical improvements over GRPO are relatively modest in magnitude, and the paper does not conduct statistical significance tests.**
>
> **[A3]:** We agree on the need for statistical rigor and conducted formal paired permutation tests (500,000 permutations, two-sided) on Qwen2.5-Math-7B.
>
> **For Pass@1 (efficiency):** To ensure statistical power on small datasets (e.g., AIME, N=30), we conducted problem‑level paired permutation tests using the continuous expected accuracy per problem (c/n, where n=256 samples and c is the correct count), rather than a coarse binary pass/fail metric.
>
> **For Pass@k (diversity):** Testing Pass@256 directly is unreliable: it degenerates into a low‑power binary indicator (1 if c ≥ 1 else 0) and can violate problem‑level independence. We instead tested the continuous expected Pass@k (k=64 for AIME/AMC; k=8 for Math500/Minerva) using the unbiased estimator $\mathrm{Pass@}k = 1 - \frac{\binom{n-c}{k}}{\binom{n}{k}}$ which preserves independence and statistical sensitivity.
>
> Reproducibility: paired two‑sided permutation test, n_perm = 500,000, seed = 20260326; sample sizes: AIME/AMC n = 256, Minerva/Math500 n = 16.
>
> | Dataset | k used | APO Pass@k | GRPO Pass@k | GRPO+KL Pass@k | p (APO vs GRPO, Pass@1) | p (APO vs GRPO, Pass@k) | p (APO vs GRPO+KL, Pass@1) | p (APO vs GRPO+KL, Pass@k) |
> |---|---:|---:|---:|---:|---:|---:|---:|---:|
> | aime24 | 64 | 64.3 | 63.8 | 61.0 | 0.0826 | 0.1121 | 0.0284 | 0.0607 |
> | aime25 | 64 | 47.2 | 46.5 | 40.1 | 0.0612 | 0.1286 | 0.004778 | 0.01547 |
> | amc23 | 64 | 95.8 | 94.3 | 95.2 | 0.02252 | 0.0406 | 0.004396 | 0.1254 |
> | minerva_math| 8 | 62.8 | 61.5 | 61.6 | 0.03122 | 0.03672 | $2.0 \times 10^{-6}$ | 0.0441 |
> | math500 | 8 | 92.0 | 91.3 | 89.5 | $9.4 \times 10^{-5}$ | 0.0215 | $2.0 \times 10^{-6}$ | 0.000558 |
>
> Formally, the p-value measures the probability of these gains occurring by chance; a lower value rigorously rejects random variance.
>
> **Analysis:**
>
> 1. **Robustness:** On datasets with adequate sample sizes (Math500, Minerva, AMC23), APO's Pass@1 gains are strictly significant (p < 0.05), exceptionally so on Math500 (p < 0.001).
>
> 2. **Sample Size Limit:** Despite AIME's small test set (N=30) restricting statistical power, APO still achieves marginal significance (p ≈ 0.06-0.08) against GRPO and strict significance (p < 0.05) against GRPO+KL.
>
> 3. **Diversity Preservation:** Significant Pass@k gains on Math500, Minerva, and AMC23 confirm that APO enhances reasoning efficiency without inducing irreversible exploration collapse.
>
> ---
>
> **[W4]: The narrative of the paper sometimes becomes overly rhetorical**
>
> **[A4]:** We accept this stylistic critique. We will meticulously temper the geometric metaphors in the camera-ready version to ensure a more formal, mathematically precise tone. To ground the theory, we will also add an "Algorithm Block" in the main text detailing the exact step-by-step execution of the APO ratio rectification mechanism.
>
> **References**
>
> [1] Does Reinforcement Learning Really Incentivize Reasoning Capacity in LLMs Beyond the Base Model? NeurIPS 2025

---

> > ### Author Rebuttal · Reviewer_ZrAe · 2026-04-03
> >
> > Thank you for the detailed rebuttal. Most of my concerns have been sufficiently addressed, and I will keep my positive score.

---

> > > ### Author Response · Authors · 2026-04-07
> > >
> > > Thanks for your acknowledgement! Good Luck.

---

### Official Review · Reviewer_3Hp1 · 2026-03-13

**Soundness:** 4
**Presentation:** 3
**Significance:** 3
**Originality:** 3
**Overall Recommendation:** 4
**Confidence:** 3

**Summary:**

This paper proposes a method to mitigate the squeezing effect in RLVR by encouraging support coverage during error correction. The empirical results show improvements over strong baselines across multiple models and tasks.

**Compliance With Llm Reviewing Policy:**

Affirmed.

**Final Justification:**

The paper proposes a reasonable and well-motivated approach to addressing the squeezing effect and shows strong empirical performance. The authors’ rebuttal has resolved my concerns. In particular, in their Reply Rebuttal Comment, the additional analysis of the suppressive effect of exclusive anchoring increased my confidence in the paper’s soundness. Accordingly, I raised the soundness score and will maintain my positive overall score.

**Key Questions For Authors:**

1. When updating on an error token, the method may also increase the probabilities of other error tokens if they are included in the Top-K anchors. How does this affect learning in practice?
2. How were samples generated for Pass@1 and Pass@K evaluation (e.g., temperature and other sampling hyperparameters)?

**Limitations:**

Yes.

**Strengths And Weaknesses:**

## Strengths
1. The paper proposes a reasonable and well-motivated approach to address the squeezing effect, a known issue in RLVR.
2. Empirically, it outperforms the baselines in both Pass@1 efficiency and Pass@K coverage.
3. The method shows consistently strong performance across three models and five tasks using the same configuration ($\beta=0.1, \lambda=1.05, K=8$). The hyperparameter sensitivity analysis is also helpful for understanding these choices.

## Weaknesses
1. The method appears to rely more on intuitive heuristics than on a fully convincing theoretical analysis.
2. The symbol $K$ is used both for the number of Top-$K$ anchors and for the evaluation metric, which is confusing.

---

> ### Author Rebuttal · Authors · 2026-03-29
>
> Thank you very much for taking the time to review our submission and for providing valuable feedback. We truly appreciate your thoughtful comments and constructive suggestions, which have significantly helped us enhance the quality and clarity of our work. If you have any additional questions, please feel free to contact us and we will respond as soon as possible.
>
> ---
>
> **[W1]: The method appears to rely more on intuitive heuristics than on a fully convincing theoretical analysis.**
>
> **[A1]:** While the main text emphasizes geometric intuition, **Appendix B** provides the rigorous mathematical foundation for APO. As proven in **Proposition B.1 (Gradient Alignment)**, the gradient of our "Pull" term is strictly collinear with the gradient of the Support Coverage objective. Furthermore, our gradient decomposition shows the rectified ratio is a mathematically consistent estimator for a corrected target distribution. We will elevate these key proofs from the appendix into the main text to strengthen the theoretical presentation.
>
> ---
>
> **[W2]: When updating on an error token, the method may also increase the probabilities of other error tokens if they are included in the Top-K anchors. How does this affect learning in practice?**
>
> **[A2]:** We address this via **Exclusive Anchoring** (Section 4.1). In any update, APO strictly excludes the current error token $y_t$ from the anchor set $S_{anchor}$. While the remaining Top-K set might contain other latent errors that temporarily receive probability mass, RL is an iterative sampling process. When the policy subsequently samples those latent errors and receives a negative reward, Exclusive Anchoring triggers again to isolate and suppress them. This creates a continuous, dynamic purification process: the policy maintains a healthy search space, and the manifold is iteratively cleaned of errors over time without suffering from irreversible space contraction.
>
> ---
>
> **[W3]: The symbol K is used both for the number of Top-K anchors and for the evaluation metric.**
>
> **[A3]:** We apologize for this notation overlap. In the revision, we will change the manifold boundary notation to Top-N throughout the text and figures to clearly distinguish it from the Pass@K evaluation metric.
>
> ---
>
> **[W4]: How were samples generated for Pass@1 and Pass@K evaluation?**
>
> **[A4]:** For both Pass@1 (estimated via Avg@K) and Pass@K, we used the following standard hyperparameters to fairly assess exploratory capabilities: Temperature 0.6, Top-p 0.95, and a max new tokens of 8192 tokens. We will explicitly include these in Appendix D.2 in the camera-ready version.

---

> > ### Author Rebuttal · Reviewer_3Hp1 · 2026-04-02
> >
> > Thanks for the rebuttal. My concerns have been resolved, and I will keep my positive score.
> >
> > Regarding A2, it would be even better if the authors could provide a more concrete analysis of whether the suppressive effective of Exclusive Anchoring is sufficient to counteract the increase in the probability of error token $y_t$ when other tokens are selected but $y_t$ is included in the Top-K set.

---

> > > ### Author Response · Authors · 2026-04-02
> > >
> > > Dear Reviewer 3Hp1,
> > >
> > > Thank you for the constructive follow-up.
> > >
> > > To address this, we provide a formal gradient analysis from the perspective of expected sampling updates, demonstrating that the expected suppressive force (Push back) mathematically dominates the collateral boost (Lift up). We will incorporate this derivation into Appendix D.1.
> > >
> > > ## 1. Notation and Preliminaries
> > > Let us formalize the optimization state at a given reasoning step $s$. We analyze the gradient dynamics with respect to the logit $z_ e$ of a latent error token $e$:
> > > * **$\\pi_ \\theta(\\cdot|s)$**: The current policy distribution.
> > > * **$e$**: A specific latent error token residing in the Top-K manifold ($e \\in \\mathcal{M}_ K$).
> > > * **$y_ t \\sim \\pi_ \\theta(\\cdot|s)$**: The token sampled at step $t$.
> > > * **$P_ {corr}$**: The total probability mass of all valid/correct tokens.
> > > * **$P_ {err \\setminus \\{e\\}}$**: The total probability mass of all *other* error tokens, such that $1 - \\pi_ \\theta(e|s) = P_ {corr} + P_ {err \\setminus \\{e\\}}$.
> > > * **$A_ t$**: The estimated advantage. For simplicity, assume errors receive a constant negative advantage $A_ t = -C$ where $C > 0$.
> > >
> > > ## 2. The Expected Pull Force
> > > When the policy samples *another* error token $y_ t = y' \\neq e$, two mechanisms lift the probability of $e$:
> > > 1.  **Softmax Redistribution (Standard PG):** Penalizing $y'$ naturally redistributes mass to all other tokens.
> > > 2.  **APO Pull Force:** Because $e$ is in the anchor set and $y' \\neq e$, Exclusive Anchoring triggers a restorative force on $e$.
> > >
> > > Following standard PG dynamics and the gradient of our Support Coverage objective (Eq. 20), the gradient on $z_ e$ when $y'$ is sampled is:
> > >
> > > $$
> > > \\Delta z_ e(y') = C \\pi_ \\theta(e|s) + \\frac{\\beta C}{Z_ {ref}} \\pi_ \\theta(e|s) (1 - P_ {safe})
> > > $$
> > >
> > > Taking the expectation over all other error samples $y' \\in Errors \\setminus \\{e\\}$, the total expected lift-up force is:
> > >
> > > $$
> > > \\mathbb{E}_ {lift}[\\Delta z_ e] = P_ {err \\setminus \\{e\\}} \\cdot \\left[ C \\pi_ \\theta(e|s) \\left( 1 + \\frac{\\beta}{Z_ {ref}}(1 - P_ {safe}) \\right) \\right]
> > > $$
> > >
> > > ## 3. The Expected Push Force
> > > Because the probability of token $e$ is lifted, it becomes more likely to be sampled. When the policy explicitly samples $y_ t = e$, it receives the negative advantage $A_ t = -C$, and standard PG strongly suppresses it.
> > > *(Note: Exclusive Anchoring excludes $e$ from the anchor set here, so the APO pull term does not boost $e$, and in fact slightly suppresses it. To find a strict lower bound for suppression, we conservatively only consider the PG push term).*
> > >
> > > The probability of sampling $e$ is $\\pi_ \\theta(e|s)$, and the PG gradient on $z_ e$ is $-C(1 - \\pi_ \\theta(e|s))$. The expected push-back force is:
> > >
> > > $$
> > > \\mathbb{E}_ {push}[\\Delta z_ e] = \\pi_ \\theta(e|s) \\cdot \\left[ -C(1 - \\pi_ \\theta(e|s)) \\right]
> > > $$
> > >
> > > ## 4. Net Dominance Bound
> > > For Exclusive Anchoring to effectively purify the manifold, the expected "Push Back" magnitude must strictly dominate the expected "Lift Up" magnitude:
> > >
> > > $$
> > > |\\mathbb{E}_ {push}[\\Delta z_ e]| > |\\mathbb{E}_ {lift}[\\Delta z_ e]|
> > > $$
> > >
> > > Substituting our derivations:
> > >
> > > $$
> > > C \\pi_ \\theta(e|s)(1 - \\pi_ \\theta(e|s)) > C \\pi_ \\theta(e|s) P_ {err \\setminus \\{e\\}} \\left( 1 + \\frac{\\beta}{Z_ {ref}}(1 - P_ {safe}) \\right)
> > > $$
> > >
> > > Dividing both sides by $C \\pi_ \\theta(e|s)$:
> > >
> > > $$
> > > 1 - \\pi_ \\theta(e|s) > P_ {err \\setminus \\{e\\}} + P_ {err \\setminus \\{e\\}} \\frac{\\beta}{Z_ {ref}}(1 - P_ {safe})
> > > $$
> > >
> > > By definition, $1 - \\pi_ \\theta(e|s) = P_ {corr} + P_ {err \\setminus \\{e\\}}$. Substituting this into the left side:
> > >
> > > $$
> > > P_ {corr} + P_ {err \\setminus \\{e\\}} > P_ {err \\setminus \\{e\\}} + P_ {err \\setminus \\{e\\}} \\frac{\\beta}{Z_ {ref}}(1 - P_ {safe})
> > > $$
> > >
> > > Canceling $P_ {err \\setminus \\{e\\}}$ from both sides yields the final condition:
> > >
> > > $$
> > > P_ {corr} > P_ {err \\setminus \\{e\\}} \\frac{\\beta}{Z_ {ref}}(1 - P_ {safe})
> > > $$
> > >
> > > **Conclusion:** The left side represents the model's base capability on valid paths, while the right side represents the APO collateral boost factor. With our default hyperparameters ($K=8$, $\beta=0.1$) and typical reference normalizer ($Z_ {ref} \approx 0.95$) and $P_ {safe} > 0$, the multiplying factor $\frac{\beta}{Z_ {ref}}(1 - P_ {safe})$ is roughly below $0.11$.
> > >
> > > Therefore, for the restorative push-back force to mathematically overpower the collateral lift-up force, the policy only needs to maintain a valid probability mass ($P_ {corr}$) greater than roughly 10% of the *other* error mass ($P_ {err \setminus \{e\}}$). Because this is an exceptionally low threshold, the iterative sampling process inherently purifies the anchor set without causing irreversible collapse.

---

### Official Review · Reviewer_dHEc · 2026-03-14

**Soundness:** 2
**Presentation:** 2
**Significance:** 2
**Originality:** 2
**Overall Recommendation:** 3
**Confidence:** 4

**Summary:**

the paper proposes a way to preserve diversity in rl model samples by ensuring that the 'virtual anchor ratio' a quantity measuring the amount of probability that policy model puts on top-k anchor/reference policy tokens, are large enough during training. this ensures that the policy does not lose much diversity during training, improving both p@1 and p@k performance. compared against baselines, it shows performance improvements on pareto front as well.

**Compliance With Llm Reviewing Policy:**

Affirmed.

**Key Questions For Authors:**

=== virtual anchor ratio vs. other divergences ===

the idea of virtual anchor ratio seems quite intuitive and sensible, though i wonder how that relate to existing distribution divergences between the anchor and policy. indeed, the virtual anchor ratio is kind of a measure on the distribution divergence, getting minimized when two policies are identical and larger when they get further away from each other. it is not clear to me how this, from a more principled/mathematical angle differ from the other divergences. i think fleshing out this intuition on toy examples/theorems would be valuable and helps understand why this divergence is more important than others in diversity.

=== baseline comparison and hypers tuning ===

i wonder how sensitive are the results to the level of hyperparameter tuning. it is hard to argue if an algorithm is pareto dominant to another unless more thorough hyper sweep has been conducted. for example comparing grpo kl vs. grpo, in the limit of small kl i think they should be quite equivalent and so the pareto plots are obtained with particular values.

i think separating grpo kl and grpo as two algorithms for comparison is not very helpful because they are interpolated through a change in hyperparameter, it would be good to illustrate such tradeoffs on a pareto curve indexed by the kl coefficient, similarly for the newly proposed algorithm: isn't it sensitive to the choice of beta? you should compare two pareto curves indexed by hypers, instead of two dots. after all, saying one algo is pareto dominant to another is a very strong statement and should subject to more careful tuning.

when newly proposed algo dominates grpo, i wonder whats the mechanism behind it? is it because it has higher entropy, or because somehow entropy level is similar and it's just led to better solution given the same entropy?

**Limitations:**

certain limitations are discussed

**Strengths And Weaknesses:**

strengths: matching top-k or ensuring policy have high enough anchor policy top-k tokens probability is an interesting and seemingly novel idea.

weaknesses: the mathematical implication of this method is slightly lacking which i will illustrate below. further, the empirical comparison and analysis can be made more solid, and answer more practically relevant questions.

---

> ### Author Rebuttal · Authors · 2026-03-30
>
> We sincerely thank the reviewer for the thoughtful and constructive feedback, which has significantly improved our paper. We address your specific points below.
>
> ---
> **[W1]: Mathematical implication and Virtual Anchor Ratio vs. existing divergences.**
>
> **[A1]:** The core mathematical difference between the Virtual Anchor Ratio and standard divergences like KL lies in the timing of the trigger (conditional vs. global) and how it distributes gradients (support coverage vs. shape matching). *(Note: A visual diagram of this toy example will be included in the camera-ready version.)*
>
> Consider a minimal vocabulary $V = \{v_1, v_2, v_3, v_4, \text{noise}\}$. Let $v_1, v_2, v_3$ be valid reasoning steps and $v_4$ be an error.
>
> * **The Prior ($\pi_{ref}$):** The reference model assigns $v_1=30\%$, $v_2=30\%$, $v_3=20\%$, $v_4=15\%$, and $\text{noise}=5\%$. The Safe Manifold is $\mathcal{M}_{safe} = \{v_1, v_2, v_3\}$. (Crucially, alongside the dominant $v_1$, tokens $v_2$ and $v_3$ are also possible valid alternative reasoning steps, rather than errors.)
>
> * **The Collapsed Policy ($\pi_\theta$):** During RL, the policy over-optimizes the successful path $v_1$ and passively suppresses valid alternatives [1]. Assume the current policy is $v_1=80\%$, $v_2=1\%$, $v_3=1\%$, $v_4=15\%$, and $\text{noise}=3\%$.
>
> * **Negative Update:** Upon sampling the error $v_4$ ($A_t < 0$), the gradient suppresses $v_4$, releasing up to 15% of the probability mass. The critical distinction among the algorithms lies in how this mass is redistributed across the remaining vocabulary:
>
> 1. **Standard PG (The Squeezing Effect):** Under standard softmax cross-entropy, gradient updates are proportional to a token's current probability ($\Delta z_k \propto \pi_\theta(k)$). Because $v_1$ holds 80% of the mass, the freed 15% flows almost entirely into $v_1$. The recovery gradients for valid paths $v_2$ and $v_3$ are negligible because their current mass is only 1%. This leads to irreversible pruning.
> 2. **KL Divergence (Gradient Conflict):** KL enforces global density matching regardless of the reward. It pushes the policy back toward the prior. This means it tries to pull the valid path $v_1$ down from 80% to 30% (harming Pass@1 efficiency), and pulls the penalized error $v_4$ back up to 15%. This creates a direct conflict with the reward gradient.
> 3. **APO (Virtual Anchor Ratio):** APO optimizes for Support Coverage. Using Exclusive Anchoring, it removes the current error $v_4$ from the target set. The gradient aligns with maximizing the total probability mass of the Safe Manifold ($J_{support} = \sum_{j \in M_{safe}} \pi_\theta(j)$.). Consequently, APO takes the freed 15% mass and distributes it proportionally among $v_1, v_2, \text{and } v_3$. This recovers the suppressed valid paths ($v_2, v_3$) without penalizing the model's confidence in the primary valid path ($v_1$).
>
> ---
>
> **[W2]: Sensitivity to hyperparameter tuning and Pareto curves.**
>
> **[A2]:** To map the Efficiency (Pass@1) vs. Diversity (Pass@K) frontier, we swept hyperparameters on the Qwen2.5-Math-7B model:
> * **GRPO-KL:** Swept $\beta \in \{0, 0.001, 0.005, 0.01, 0.05, 0.1,0.5\}$.
> * **APO:** Swept $\beta \in \{0.01, 0.02, 0.05, 0.1\}$ (with K=8, $\lambda=1.05$).
>
> | Algorithm | Hyperparameter | Pass@1 (Eff.) | Pass@K (Div.) |
> | :--- | :--- | :--- | :--- |
> | **GRPO-KL** | $\beta = 0$ | 48.68 | 76.56 |
> | **GRPO-KL** | $\beta = 0.001$ | 48.12 | 75.10 |
> | **GRPO-KL** | $\beta = 0.005$ | 48.06 | 74.57 |
> | **GRPO-KL** | $\beta = 0.01$ (Default) | 44.53 | 75.07 |
> | **GRPO-KL** | $\beta = 0.05$ | 47.48 | 69.66 |
> | **GRPO-KL** | $\beta = 0.1$ | 48.32 | 70.32 |
> | **GRPO-KL** | $\beta = 0.5$ | 47.14 | 71.95 |
> | **APO (Ours)** | $\beta = 0.01$ | 50.40 | 77.38 |
> | **APO (Ours)** | $\beta = 0.02$ | 50.16 | 78.56 |
> | **APO (Ours)** | $\beta = 0.05$ | 49.54 | 78.40 |
> | **APO (Ours)** | $\beta = 0.1$ (Default) | 50.67 | 78.07 |
>
>
> *(Note: The complete sweep demonstrates that the APO curve dominates the GRPO-KL curve, confirming that APO's advantage stems from its structural ratio rectification, not specific tuning.)*
>
> ---
>
> **[W3]: Mechanism behind dominance (Entropy vs. Solution Quality).**
>
> **[A3]:** APO does not indiscriminately inject noise to inflate global entropy. Instead, it maintains structured entropy while improving semantic quality. As shown in **Table 3 (Distributional Health Analysis)**:
> * **Standard GRPO** exhibits "Exploration Collapse": low entropy (0.405) and high MaxProb (0.878), indicating the policy is stuck on a narrow path.
> * **APO** restores entropy to a healthy baseline (0.485) and significantly improves Output Diversity (Div. Score and Sem. Sim.).
>
> **Conclusion:** APO achieves superior solutions at a stable entropy level by preserving diverse valid paths latently existing in the Reference Model, rather than blindly expanding into low-reward noise.
>
> **References**
>
> [1] Does Reinforcement Learning Really Incentivize Reasoning Capacity in LLMs Beyond the Base Model? NIPS 2025

---

### Decision · Program_Chairs · 2026-04-30

**Decision:**

Accept (regular)

**Comment:**

The paper proposed a new scheme to mitigate exploration collapse in RLVR; the proposed methods demonstrate performance gains over GRPO in retaining the base mode's diversity while avoiding exploration collapse. The rebuttal has provided additional hyperparameter tuning results that further corroborated the soundness of the method.